# Non-neutralizing antibodies elicited by recombinant Lassa–Rabies vaccine are critical for protection against Lassa fever

Tiago Abreu-Mota [1,2,3], Katie R. Hagen[4], Kurt Cooper[4], Peter B. Jahrling[4,5], Gene Tan[6,7], Christoph Wirblich[1], Reed F. Johnson[5] & Matthias J. Schnell[1,8]

Lassa fever (LF), caused by Lassa virus (LASV), is a viral hemorrhagic fever for which no approved vaccine or potent antiviral treatment is available. LF is a WHO priority disease and, together with rabies, a major health burden in West Africa. Here we present the development and characterization of an inactivated recombinant LASV and rabies vaccine candidate (LASSARAB) that expresses a codon-optimized LASV glycoprotein (coGPC) and is adjuvanted by a TLR-4 agonist (GLA-SE). LASSARAB elicits lasting humoral response against LASV and RABV in both mouse and guinea pig models, and it protects both guinea pigs and mice against LF. We also demonstrate a previously unexplored role for non-neutralizing LASV GPC-specific antibodies as a major mechanism of protection by LASSARAB against LF through antibody-dependent cellular functions. Overall, these findings demonstrate an effective inactivated LF vaccine and elucidate a novel humoral correlate of protection for LF.

[1] Department of Microbiology and Immunology, Sidney Kimmel Medical College at Thomas Jefferson University, Philadelphia, PA 19107, USA. [2] Life and Health Sciences Research Institute (ICVS), School of Medicine, University of Minho, 4710-057 Braga, Portugal. [3] ICVS/3B's, PT Government Associate Laboratory, Braga/Guimarães, 4710-057, Portugal. [4] Integrated Research Facility, National Institute of Allergy and Infectious Diseases, National Institutes of Health, Fort Detrick MD 21702, USA. [5] Emerging Viral Pathogens Section, National Institute of Allergy and Infectious Diseases, National Institutes of Health, Bethesda MD 20892, USA. [6] Infectious Disease, The J. Craig Venter Institute, La Jolla, CA 92037, USA. [7] Department of Medicine, University of California, San Diego, La Jolla CA 92037, USA. [8] Jefferson Vaccine Center, Sidney Kimmel Medical College at Thomas Jefferson University, Philadelphia, PA 19107, USA. Correspondence and requests for materials should be addressed to M.J.S. (email: Matthias.schnell@jefferson.edu)

Lassa fever (LF) is a viral hemorrhagic fever (VHF) whose etiologic agent is Lassa virus (LASV), a bio-safety level 4 (BSL-4) pathogen. Similar to other VHFs caused by other viruses, such as Ebola virus (EBOV) and Marburg virus (MARV), LF can be highly fatal and no vaccine is currently available[1]. The need to develop vaccines against emerging viral pathogens became starkly apparent during the 2014–2016 West Africa Ebola epidemic[2–4]. Indeed, reaffirming the urgency and importance of preventive measures, an unprecedented major LF surge, with 25.4% high case fatality rate, is currently unfolding in Nigeria[5]. Unlike most other BSL-4 agents which cause temporally and geographically confined epidemics, LF is believed to be widespread throughout most of West Africa, with an estimated 100,000–300,000 humans infected annually[6,7]. As many as 80% of LF exposures are mildly symptomatic and thus go unreported[6], however, the case fatality rate of LF has been reported to reach as high as 50%[8]. Such discrepancy can be dependent on both the contributing strain and the population afflicted (e.g., pregnant women are especially susceptible)[9,10]. Even among survivors, LF can cause severe neurosensory sequela; it is a leading cause of viral-induced neurosensory deafness in West Africa[8].

A logistical hurdle for an effective LASV treatment is the often poorly equipped health infrastructure in developing nations such as Guinea or Sierra Leone[11]. While the off-label use of ribavirin seems effective in treating LF, the drug is often accompanied by severe side effects. Coupled with the presence of conflict-stricken regions, the relative remoteness of some human settlements and the widespread presence of LASV's natural reservoir, *Mastomys natalensis* (common African rat), both diagnosis and treatment of LF is a challenging task[12]. With climate change and increasing globalization, the likelihood of LF becoming a global threat increases, thus making development of a vaccine for LASV a high priority.

Unfortunately, undefined correlates of protection for LF have impeded LASV vaccine development. Studies with experimental live vaccines, such as ML29 (a Mopeia–Lassa virus reassortment-based vaccine) and recombinant vaccina virus expressing LASV glycoproteins, have shown that cellular immunity occurs in the absence of humoral response and successfully protects treated animal model[13,14]. Additionally, these findings, together with findings on another promising LASV vaccine platform, VSV-LASV, have indicated that either no correlation, or even a negative correlation, exists between LASV humoral response and vaccine efficacy[15,16]. Nevertheless, it has also been shown in some animal models that cellular immunity may be the source of immunopathology seen in LF[17–19]. Meanwhile, studies have reported that passive sera transfer therapy from LF survivors protects against disease and death in animal models of LF, supporting the role of humoral response against disease development[20,21].

LASV's genome, as a member of the Arenaviridae family, encodes four proteins, including an envelope glycoprotein that is responsible for viral entry[22]. LASV's glycoprotein is expressed as a polyprotein and is cleaved into SSP, GP1, and GP2 to form a mature trimeric glycoprotein complex (GPC) on the surface of host cells and virions[22]. GPC is an appealing immunogen because of the surface exposure of GPC in LASV virions and its crucial function for viral entry[16,23–26]. Indeed, human monoclonal antibodies that target GPC and neutralize LASV in vitro were recently shown to protect guinea pigs and non-human primates (NHPs) exposed to LASV from disease[25,27]. However, the efficacy of GPC-specific non-neutralizing mAbs was not investigated and neutralizing potency in vitro did not necessarily correlate with protection[25,26]. Furthermore, the occurrence of neutralizing antibodies (NAbs) against LASV is uncommon in survivors and has been poorly elicited by previous LASV vaccine strategies[28].

Besides direct viral neutralization, antibodies can also lead to effector cell activation and clearance of the viral antigen-expressing cells through antibody-dependent cellular cytotoxicity (ADCC) or phagocytosis (ADCP)[29]. Through this mechanism, antibodies bound to antigen interact with Fcγ-receptor-bearing immune effector cells, such as macrophages or NK cells, through Fc region cross-linking[29] that triggers clearance of the antigen-expressing cell. As such, ADCC/ADCP are among several mechanisms that bridge the adaptive and innate immune responses. ADCC/ADCP has been shown to be highly relevant for protecting against and clearing several different viruses, including HIV, influenza virus, and EBOV[30–34]. However, the role of ADCC, ADCP, and other antibody-mediated effector functions in LASV infection and disease outcome has not been investigated.

Here we report the use of a rabies virus (RABV)-based vaccine vector as an inactivated dual vaccine for LASV and RABV. This vaccine, named LASSARAB, expresses a codon-optimized version of LASV GPC (coGPC) in addition to RABV G. LASSARAB elicits lasting humoral response against LASV and RABV in both mouse and guinea pig models, and it protects both against LF. In developing LASSARAB, we also sought to uncover its mechanism of protection, which our results suggest is dependent on a previously uncharacterized antibody-mediated protection of LASV through effector cell functions of GPC-targeted non-neutralizing antibodies (Non-NAbs).

## Results

**Generation of rhabdoviral-based vectors expressing LASV GPC.** To generate a recombinant RABV-expressing LASV GPC, we used the previously described vector BNSP333[35]. BNSP333 is a modified RABV vaccine strain (SAD B19) with an arginine-to-glutamate change at position 333 of RABV G that further reduces neurotropism and improves its safety profile[35]. A codon-optimized LASV-GPC was cloned into BNSP333 using two unique restriction sites (BsiWI and NheI) that flank a RABV transcription start/stop signal between the RABV N and P genes, and it was designated as LASSARAB (Fig. 1). Utilizing LASSARAB, we also constructed LASSARAB-ΔG by deleting the RABV G. For a control vector, we constructed a recombinant vesicular stomatitis virus (VSV) expressing the same GPC as the RABV vector (rVSV-GPC); similar to LASSARAB-ΔG, it lacks its native glycoprotein (G). In several prior NHP studies, similar rVSV-GPC vectors have been used as live-attenuated (replication-competent) vaccine candidates for LASV with promising results[15,16]. As an additional control, we used BNSP333-expressing Ebola GP (FILORAB1), a vaccine extensively characterized by our group[36–38].

**GPC is transported to the cell surface and incorporated into virions.** Successful utilization of LASSARAB and LASSARAB-ΔG as vaccines depends on LASV GPC expression at the cell surface membrane. VERO cells were infected at a multiplicity of infection (MOI) of 0.1 or 1, and cell surface expression of LASV GPC and RABV G was evaluated by immunofluorescence and flow cytometry at 48 h post-infection (Figs 2a, b). Immunostaining with antibodies directed against either LASV GPC or RABV G detected both LASV GPC and RABV G cells on the cellular surface of VERO cells infected with LASSARAB (Figs 2a, b panel LASSARAB). In cells infected with FILORAB1, only RABV G was detected on the cell surface as expected (Figs 2a, b, panel FILORAB1) whereas for the LASSARAB-ΔG and rVSV-GPC-infected cells, LASV GPC but not RABV G was detected on the cell surface (Fig. 2b panel LASSARAB-ΔG/rVSV-GPC).

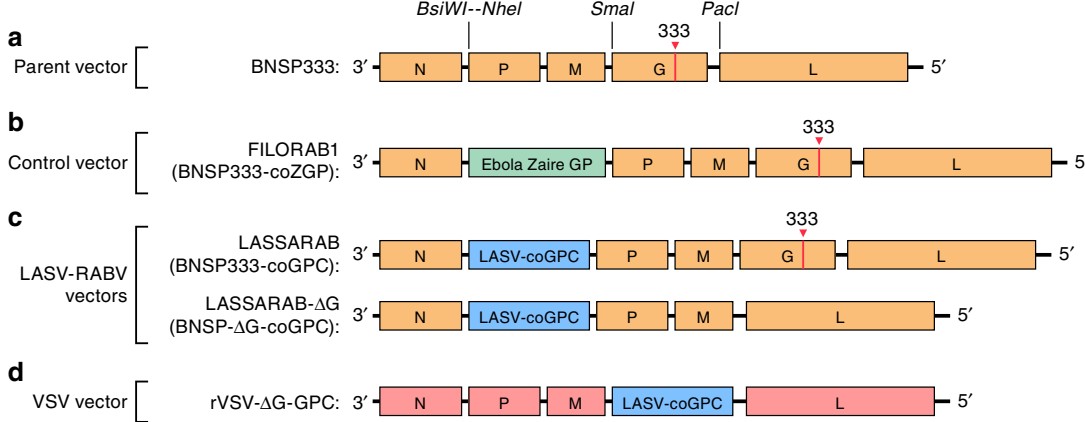

**Fig. 1** Diagram of vaccine constructs and controls. BNSP333 is the parental vector and FILORAB1, the control used, is based on BNSP333 with a codon-optimized Zaire Ebola Virus Glycoprotein (EBOV GP) inserted between N and P through the BsiWI and NheI restriction digest sites. LASSARAB was generated in a similar manner as FILORAB1, from BNSP333 by cloning a codon-optimized version of Lassa virus glycoprotein (LASV GPC) in the BsiWI and NheI restriction digest sites. LASSARAB-ΔG was further generated from LASSARAB by removing the native rabies glycoprotein (G) by using the restriction digest sites SmaI and PacI. rVSV-GPC was generated by replacing the native VSV glycoprotein (G) by LASV GPC at the same sites. rVSV-GPC was created to be used as a control vector and as a scaffold to produce a native LASV GPC antigen for ELISAs (see Methods section)

To analyze whether LASV GPC affects RABV growth kinetics, we performed a multi-step growth curve analysis of LASSARAB, LASSARAB-ΔG, and FILORAB1 (Fig. 2c). LASSARAB and FILORAB1 grew similarly and reached titers of $10^8$ after 72 h. LASSARAB-ΔG grew to a higher titer than the RABV G-containing construct LASSARAB, indicating that LASV GPC is being functionally expressed. The higher titer achieved by LASSARAB-ΔG might be explained by its shorter genome or its expression of two glycoproteins, or both.

LASSARAB's potential as an inactivated vaccine depends on LASV GPC incorporation in LASSARAB-inactivated virions. As such, sucrose-purified virions from infected VERO cells were analyzed by SDS-PAGE gel, western blotting, and ELISA (Figs 2d, e, and Supplementary Fig. 1). SDS-PAGE protein stain of purified FILORAB1 (control) and LASSARAB virions showed protein migration in the expected size for the RABV proteins, as can be seen by the FILORAB1 control, as well as proteins consistent with the molecular weight of LASV GP2 (40–38 kDa). LASV GP1 (47–42 kDa) is comigration with RABV P and therefore difficult to detect. However, LASV GP1/GP2 incorporation in LASSARAB was confirmed by western blot analysis which demonstrated both GP1 (48–42 kDa) and GP2 (40–38 kDa) consistent with their respective molecular sizes (Fig. 2e and Supplementary Fig. 1)[39–41]. Glycosylation patterns in both GP1 and GP2 similar to previous studies were demonstrated by mobility shift assay using LASSARAB virions treated with either Endo H or PNGase F in comparison with untreated virions (Supplementary Fig. 1e and f)[39–41]. Finally, to confirm that LASV GPC on inactivated LASSARAB particles was conformationally resent in its pre-fusion state, particles were analyzed by the GPC conformational sensitive mAb 37.7H[26,42] (Supplementary Fig. 1g).

**LASSARAB is avirulent in mice.** Expression and incorporation of LASV GPC in the highly attenuated BNSP333 live vaccine vector might change its tropism and thus increase its pathogenicity. To determine whether this is the case, Swiss Webster mice were inoculated both intranasally (IN) and intraperitoneally (IP) with $10^6$ foci-forming units (ffu) of LASSARAB, LASSARAB-ΔG, FILORAB1, or $10^6$ plaque-forming units (pfu) rVSV-GPC, or PBS. Animals were monitored for disease (e.g., hunched back, ruffled fur) and changes in weight for 28 days (Fig. 3a). IN exposure with BNSP (RABV group), which has been shown to be

pathogenic after IN exposure, was used as a positive control, while FILORAB1 and PBS were used as negative controls because previous studies had demonstrated that they are not virulent[43,44]. On day 8, RABV-infected animals started to exhibit clinical signs of rabies, particularly weight loss. (Fig. 3a, RABV group). Mice inoculated with LASSARAB or FILORAB1 showed no clinical signs of disease. For the LASSARAB-ΔG IN inoculated group, one mouse died at day 14 without displaying previous clinical signs or weight loss (Fig. 3a, LASSARAB-ΔG group, m2). However, three mice from the rVSV-GPC group displayed signs of neurological deficits (Fig. 3a, rVSV-GPC group, m2/4/5); two succumbed and one survived, indicating pathogenicity after IN inoculation of this vaccine. None of the animals inoculated through the IP route displayed clinical signs of disease.

We further characterized the safety profile of the infectious LASSARAB vaccine by intracranial inoculation (IC) in both adult BALB/c and adult severe combined immunodeficiency (SCID) mice (3b). Increased pathogenicity was not observed following infections with LASSARAB compared with BNSP333 in either Balb/C or SCID mice (Fig. 3b). Finally, to confirm absent or decreased pathogenicity in a more sensitive model[44], Swiss Webster suckling mice were IC-exposed with LASSARAB or BNSP333 (Fig. 3c). Independent of the virus dose used, LASSARAB or BNSP333 suckling mice started to succumb to the infection by day 7.

**Live LASSARAB does not induce LASV-specific GPC IgGs.** We first evaluated immunization with replication competent vaccines. All live-attenuated (replication-competent) RABV based vaccines will be referred from now on with an rc- suffix (e.g., rc-LASSARAB). rVSV-GPC is always used as replication competent vaccine. C57BL/6 mice were intramuscularly immunized on day 0 with $10^6$ ffu rc-LASSARAB, rc-LASSARAB-ΔG, rc-FILORAB1, or $10^6$ pfu of rVSV-GPC. Humoral immune responses were analyzed by a newly developed LASV GPC-specific ELISA bi-weekly until day 42 post-immunization (Supplementary Fig. 1 and 2). By day 14, both rc-FILORAB1- and rc-LASSARAB-immunized mice had high titers of RABV-G-specific total IgG, and by day 28, maximum titers were achieved and were maintained until day 42, as seen previously (Supplementary Fig. 2)[36]. rc-LASSARAB-ΔG and rVSV-GPC immunized mice did not seroconvert to RABV-G. In contrast, LASV GPC-specific titers were detected in

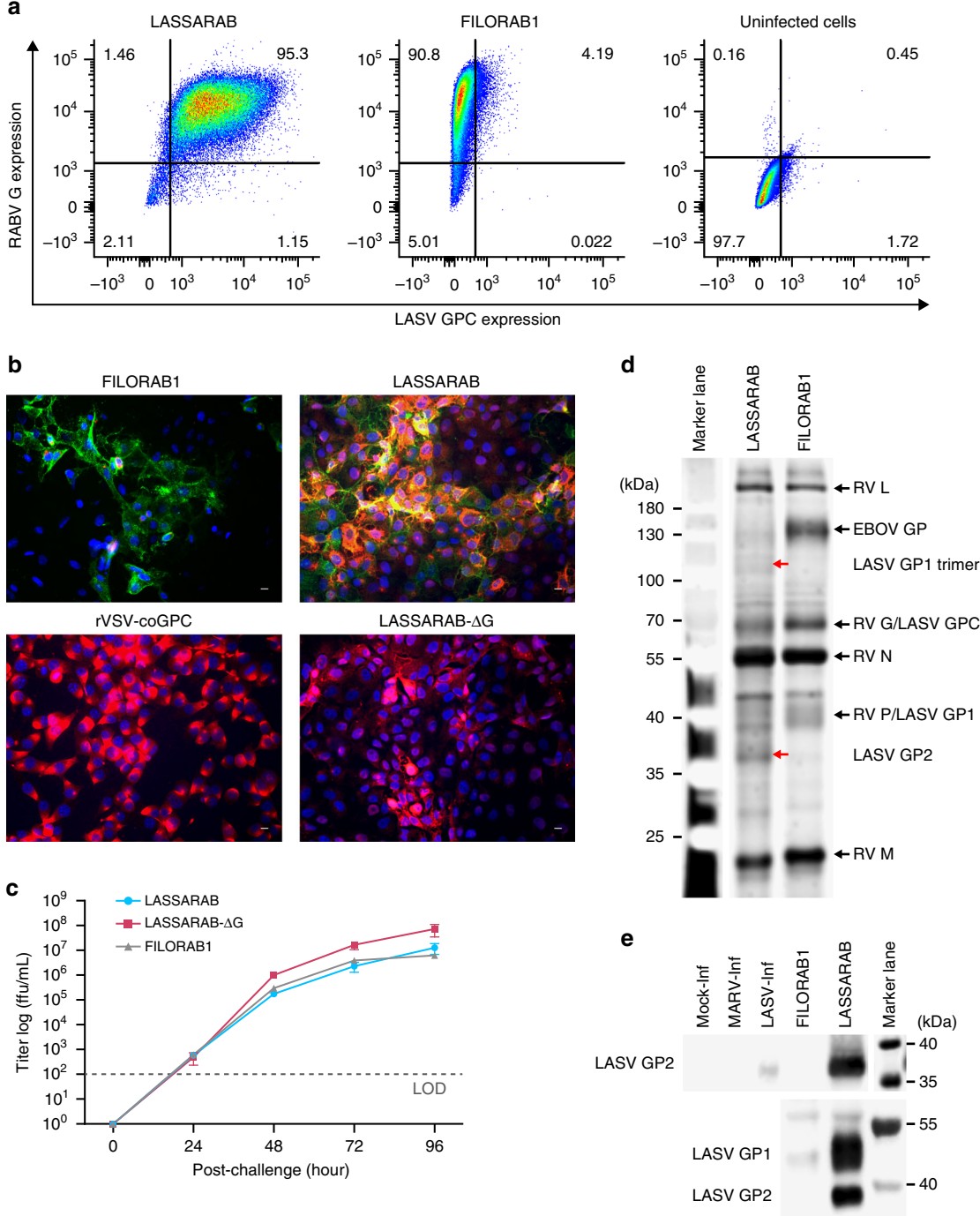

**Fig. 2** Evaluation of LASSARAB and LASSARAB-ΔG vectors in cell culture and inactivated virion characterization. **a** LASSARAB, FILORAB1 and uninfected VERO cells were probed for LASV-GPC and RABV-G expression with 37.7H anti-LASV human mAb and 1C5 anti-RABV G mouse mAb and analyzed by flow cytometry 48 h post infection. **b** VERO cells were infected at a MOI of 0.1 with 4 viruses: FILORAB1, LASSARAB, LASSARAB-ΔG, and rVSV-coGPC. 48 h later (24 h for VSV based vectors) cell surface expression of LASV glycoprotein (GPC), in red, and RABV glycoprotein (G), in green, was probed by a α-LASV GPC rabbit polyclonal and a α-RABV G human 4C12 monoclonal, respectively. In LASSARAB infected cells, yellow is observed as the superimposition of LASV GPC surface expression with RABV G. The bar indicates 12 μm. **c** VERO CCL-81 cells were infected with a MOI of 0.01 and media supernatant was collected at 0, 24, 48, 72, and 96 h. Virus titers were measured through foci-forming assay (in *Y*-axis) and plotted through time (*X*-axis). **d**, **e** LASSARAB and FILORAB1 virions were concentrated through TFF and sucrose purified through ultra-centrifugation. Pellets were resuspended in PBS, BPL inactivated at 1:2000 for 24 h, and 2 μg of each was loaded in a denaturing 10% SDS-PAGE gel. In **d** SYPRO Ruby staining was used. **e**, **f** LASV GPC incorporation in LASSARAB particles was confirmed by western Blot with either an anti-LASV GP2 rabbit polyclonal (upper panel) and anti-GPC/GP1/GP2 guinea pig survivor serum (lower panel). Uncropped versions are available in supplementary figures

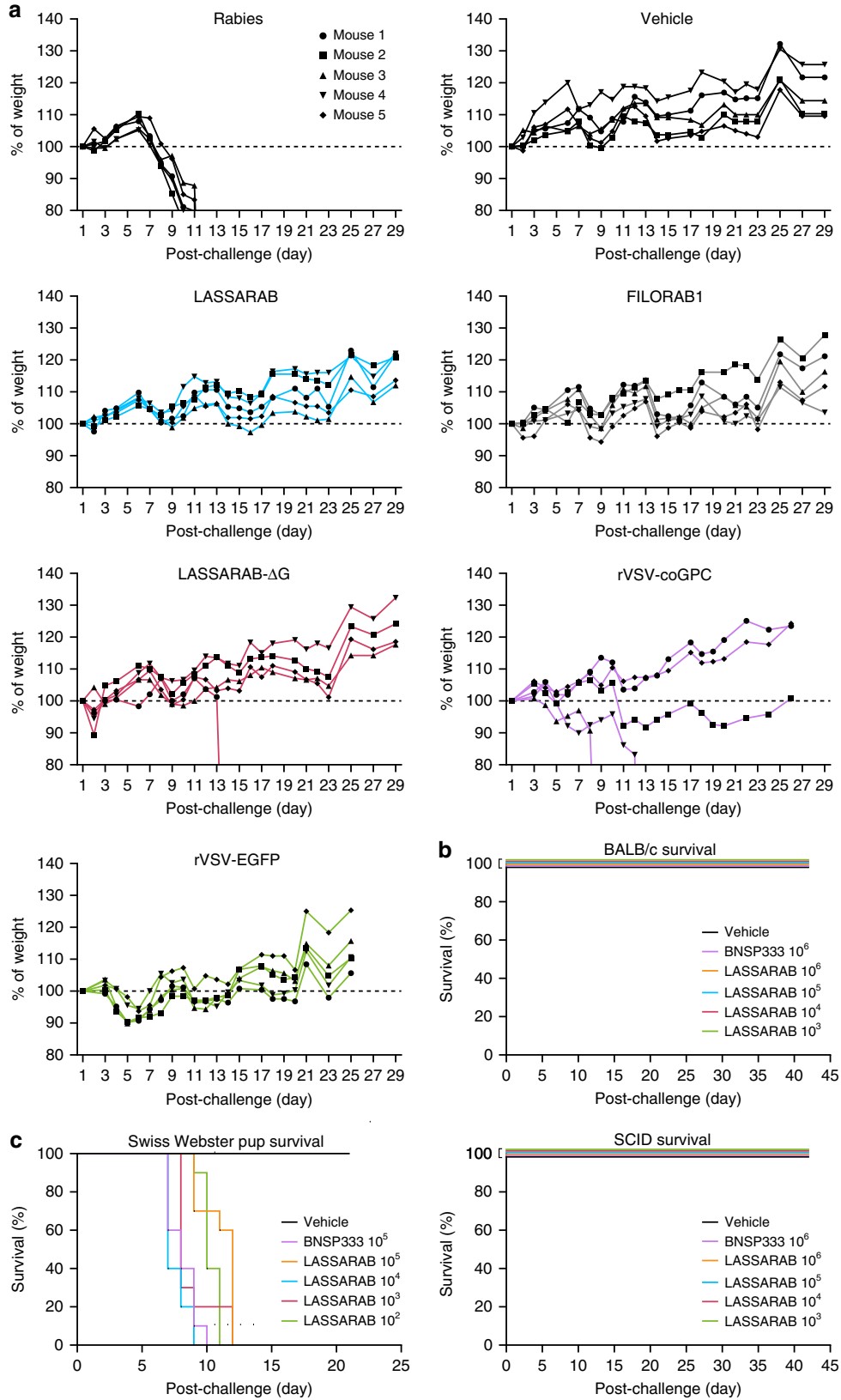

rc-LASSARAB-ΔG immunized mice only, and only at low titers on days 28 and 42 (Supplementary Fig. 2). rVSV-GPC had a significant LASV GPC-specific immune response (Fig. 4c, purple line).

**Inactivated-LASSARAB virions induce humoral response in mice**. We also explored the humoral immunogenicity of inactivated LASSARAB virions. Inactivated LASSARAB or FILORAB1 virions will simply be referred as LASSARAB or FILORAB1. We

**Fig. 3** Evaluation of LASSARAB, LASSARAB-ΔG, and rVSV-GPC pathogenicity. **a** Weight curves of 6- to 8-week-old female Swiss Webster mice that were inoculated intranasally with $10^5$ ffu of either LASSARAB, LASSARAB-ΔG, or rVSV-GPC. As controls, mice were inoculated with the same dosage of either BSNP parent vector (Rabies) without the 333 mutation in the Rabies G, FILORAB1, rVSV-EGFP, or Mock (PBS). Weight is standardized as percentage of weight loss or gain in comparison with first day of exposure. Rabies virus-infected animals developed clinical signs on day 8 with further weight loss until day 11 when endpoint criteria were reached. In LASSARAB-ΔG one mouse died at day 14 without displaying any signs or weight loss. In rVSV-GPC, three mice displayed signs of neurological deficit with two succumbing and one surviving. All other mice showed no signs of pathology. **b** Survival curves of BALB/c or SCID mice that were subjected to intracranially (IC) exposure with either LASSARAB or BNSP333. No signs of disease nor death were observed post- exposure. **c** IC exposure of Swiss Webster suckling mice with either LASSARAB or BNSP333. Suckling mice started succumbing to infection by day 7 in BNSP333 group and survived as long as day 12 in LASSARAB group with none surviving by the end of the study

intramuscularly administered 10 μg of β-propiolactone (BPL)-inactivated LASSARAB or FILORAB1 particles to C57BL/6 mice following the standard three-inoculation RABV vaccination schedule (Fig. 4a). Both vaccines were further tested in two different formulations: either in PBS only (LASSARAB/FILORAB1 groups), or adjuvanted with TLR4 receptor agonist (Glucopyranosil Lipid A) in a stable emulsion (LASSARAB+GLA-SE group)[45]. GLA-SE is a clinical-trial stage adjuvant that has been shown to enhance the breadth and quality of humoral immune responses for FILORAB1 and influenza virus[37,38,46]. Blood was collected and the humoral immune response was analyzed periodically until day 42 (Fig. 4 and Supplementary Fig. 2). Analysis of total IgG against LASV GPC by ELISA indicated seroconversion at day 14 by both LASSARAB and LASSARAB+GLA-SE groups; by day 28 both achieved statistical significance in comparison to control groups (Fig. 4b). Since endpoint titers of both inactivated LASSARAB and LASSARAB+GLA-SE had achieved appreciable total IgG responses against LASV GPC, we examined the quality of this humoral response by IgG2c and IgG1 sub-isotype-specific LASV GPC ELISA. IgG1/IgG2c ratios lower than 1.0 indicated an increasing Th1-bias response, which is desirable for antiviral responses. LASSARAB+GLA-SE not only achieved a significantly higher IgG2c response than LASSARAB, but also achieved consistently lower and uniform IgG1/IgG2c ratios (F-test, $p < 0.01$), thus decreasing the variability of the immune response between mice (Figs 4d, e).

**LASSARAB does not induce neutralizing antibodies**. The development of NAbs was investigated for LASSARAB using a pseudotyped VSV in vitro assay. This assay utilizes a single round ΔG-rVSV pseudovirus (ppVSV) which expresses both NanoLuc and eGFP as reporter genes[38,47]. When pVSV pseudotyped with RABV G was used, the sera of either replication-competent or inactivated LASSARAB achieved high NAbs against RABV G (> 10,000) compared to negative controls (Fig. 5, RABV). Since RABV G NAbs are a correlate of protection against RABV, these results indicated that LASSARAB is a suitable vaccine against RABV. Protection by RABV NAbs was further confirmed by using the WHO standard (Figs 5a, c) in which values > 0.5 IU/ml are considered protective against RABV; every group achieved IU/ml values much higher than 0.5 IU/ml, indicating that the addition of LASV GPC in the RABV backbone did not compromise its ability to generate RABV NAbs. Conversely, when ppVSV was pseudotyped with LASV GPC, we were not able to detect GPC-specific NAbs both in the presence or absence of complement (Supplementary Fig. 3), whereas the control human mAbs (12.1F, 25.10C and 37.7H) exhibited neutralizing activity at similar concentrations as described[26], indicating that our assay was functional (Figs 5a, b).

**LASSARAB+GLA-SE is efficacious in guinea pigs**. We evaluated LASSARAB vaccine efficacy using outbred Hartley guinea pigs and the guinea pig-adapted LASV[48]. Six groups of ten Hartley guinea pigs were used (Fig. 6a): three groups were

immunized with inactivated LASSARAB+GLA-SE particles once (1), twice (2), or three times (3); two groups were immunized with replication competent LASSARAB (rc-LASSARAB) or rVSV-GPC; and one group received RabAvert. All groups were challenged 58 days after the primary immunization with $10^4$ pfu of the guinea pig-adapted LASV Josiah strain. The animals were monitored for viremia and clinical signs were recorded daily up to day 47 post-challenge (Figs 6b, c). Significant protection was observed for animals immunized three times with LASSARAB+GLA-SE ($p = 0.0019$) or replication competent rVSV-GPC ($p = 0.0008$) (Fig. 6b, red and purple lines). Guinea pigs inoculated with rc-LASSARAB or immunized once or twice with LASSARAB+GLA-SE showed no significant protection but a trend toward it. Interestingly, remarkably different clinical signs were observed in the two groups that were protected against LASV exposure (Fig. 6c, rVSV-GPC&LASSARAB+GLA-SE (−58, −51, −30) groups). While all animals in rVSV-GPC vaccinated group had an onset of clinical signs by day 12, all but two of the LASSARAB+GLA-SE immunized animals were free of clinical signs of disease. Curiously, in endpoint qPCR LASV RNA viremia analysis (Fig. 6d, survivors group), ~ 20% of surviving animals across all groups (except rc-LASSARAB and RabAvert) had an average of $10^5$ LASV RNA copies per ml, indicating that despite being protected, some viremia was still present (Fig. 6d).

Next, we analyzed endpoint NAbs titers by LASV GPC pseudotyped ppVSV (Fig. 6e). The NAb response was highly variable across groups, being present in both survivors and succumbed animals with no significant difference between them ($p = 0.18$). These data indicated that either NAbs play a minor role in survival or, in the case of the succumbed animals, develop too late in the infection to play a significant role.

The absence of NAbs against LASV across survivors led us to investigate correlates of protection in surviving guinea pigs by analyzing total IgG levels against LASV GPC in both pre-challenge and post-challenge serum (Figs 6f, g). As shown in Fig. 6f, the groups that were protected against challenge, rVSV-GPC and LASSARAB+GLA-SE (3), had significantly higher titers of LASV GPC specific IgG ($p = 0.0001$ and $p < 0.0001$, respectively) in the pre-challenge sera when compared to RabAvert group. When post-challenge terminal sera were assayed (Fig. 6g), concentrations of LASV GPC-specific IgG were significantly higher in survivors compared to animals that succumbed ($p < 0.0001$). Overall, our data suggest that, in both prior and post-exposure to LASV, higher levels of non-neutralizing LASV GPC-specific IgGs correlate with protection.

**LASSARAB induced non-neutralizing antibodies stimulate ADCC**. Once we found that a high LASV GPC-specific IgG titer with low or no NAbs correlated with protection in the LASSARAB+GLA-SE group, we determined whether non-NAb can mediate protection through cell-mediated mechanisms, such as ADCC or ADCP. For this purpose, we used sera from mice immunized twice (on day 0 and day 28) with LASSARAB+GLA-SE (LASSARAB sera) or FILORAB1+GLA-SE (control)

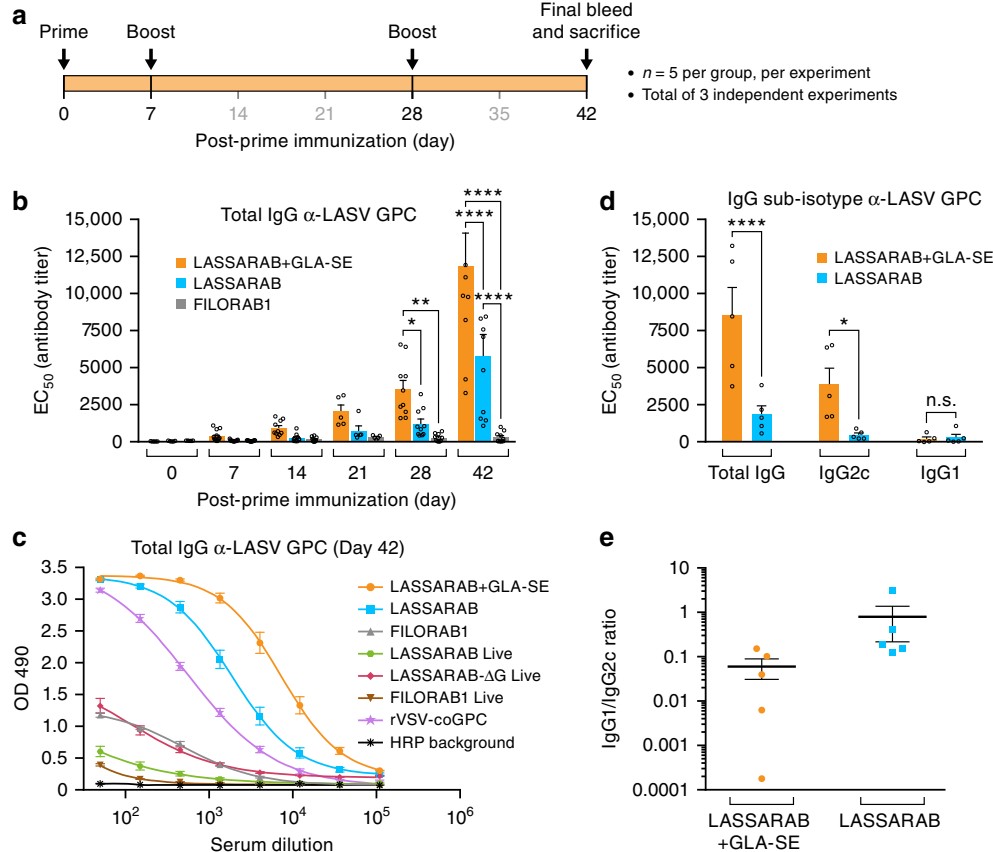

**Fig. 4** Analysis of the humoral response towards Lassa virus glycoprotein. C57BL/6 mice were immunized IM in the gastrocnemius muscle with either 10 μg of β-propiolactone inactivated viral particles in PBS or adjuvanted with 5 μg of GLA, a TLR-4 agonist formulated in 2% of stable emulsion (SE); LASSARAB+GLA-SE, LASSARAB, FILORAB1 groups) and boosted two times with the same amount on day 7 and 28 (**a**). Immunizations with replication-competent viruses were executed with a single time inoculation of 10[6] ffu or pfu virus IM in the gastrocnemius (rc-LASSARAB; rc-FILORAB1 groups and rVSV-GPC). **b** The EC$_{50}$ values (obtained from the 4PL regression ELISA curve) of the total IgG titers against LASV GPC are plotted since day 0 until day 42. Error bars are representative of the standard error mean (SEM) and is calculated from 15 mice per group. Statistical significance was calculated by using 2-way ANOVA–post-hoc Tukey's Honest Significant Difference Test. **c** ELISA of total IgG against LASV GPC of all day 42 groups are shown for all immunized groups. ELISA curves are generated from 4PL regression. Error bars are representative of the SEM of OD 490 values (five mice per group, in triplicates). **d** Day 35 EC$_{50}$ antibody titer of IgG sub-isotype (IgG2c and IgG1) against LASV GPC of sera from LASSARAB+GLA-SE and LASSARAB group was analyzed. Error bars are the SEM of a total of five mice per group and statistical significance by 2-way ANOVA (post-hoc Tukey's Honest Significant Difference Test). **e** The ratios of the respective EC$_{50}$ antibody titers IgG1/IgG2c are plotted and the $F$ test was applied to check for variance difference ($p < 0.001$). (****$P < 0.0001$; ***$P < 0.001$; **$P < 0.01$; *$P < 0.05$)

(Supplementary Fig. 3). First, we analyzed NK cell-mediated ADCC activity using an in vitro assay modified from a previously described rapid and fluorometric antibody-dependent cellular cytotoxicity (RFADCC)[49]. Briefly, we developed a stable 3T3 cell line expressing LASV GPC (3T3-LASV) and used it as target cells, and purified murine C57BL/6 NK cells as effectors, as described in Methods and Supplementary Fig. 3. 3T3-LASV cells were incubated with either LASSARAB sera or control, and different ratios of effector cells to target cells (E:T) were used (Fig. 7a and Supplementary Fig. 3). In the presence of LASSARAB sera, murine NK cells mediated significantly more killing ($p < 0.01$) at any E:T compared to controls (Fig. 7a). This effect was reduced to background levels when another 3T3-based cell line expressing an irrelevant viral glycoprotein (3T3-MARV) was used as a target cell (Supplementary Fig. 3).

To determine which antibody isotype is important for ADCC-mediated killing of 3T3-LASV, we isolated IgG from the sera and conducted the assay with 40 μg/ml of either purified IgG or IgG-depleted sera (Fig. 7b and Supplementary Fig. 3). Again, killing of 3T3-LASV was significantly higher in the presence of LASV-specific purified IgG than in the control; in contrast, target cell

cytotoxicity was reduced to background levels when IgG-depleted sera were used. Together these findings indicate that ADCC is mediated by the LASV GPC-specific IgG.

**Macrophages mediate ADCP after immunization with LAS-SARAB.** To examine whether other antibody-dependent cell-mediated mechanisms are involved in the clearance of LASV, we modified our ADCC assay to test if macrophages are involved in ADCP. As seen for the NK cells, peritoneal C57BL/6 macrophages (IC-21) induced 3T3-LASV cell killing compared to control sera when incubated with LASSARAB sera (Fig.7d and Supplementary Fig. 3). Moreover, we observed that peritoneal BALB/c macrophages (J774A.1) internalized 3T3-LASV cells in the presence of LASSARAB sera, likely through ADCP (Figs 7c, e). Target cell internalization was confirmed to be dependent upon Fcγ-R activation as macrophages incubated with anti-Fcγ-RIII mAb (but not anti-Fcγ-RI or anti-Fcγ-RIV) abolished 3T3-LASV internalization to background levels (Fig. 7c).

**Fcγ-receptor function is critical for protection in mice.** We also investigated the relevance of antibody cellular effector function

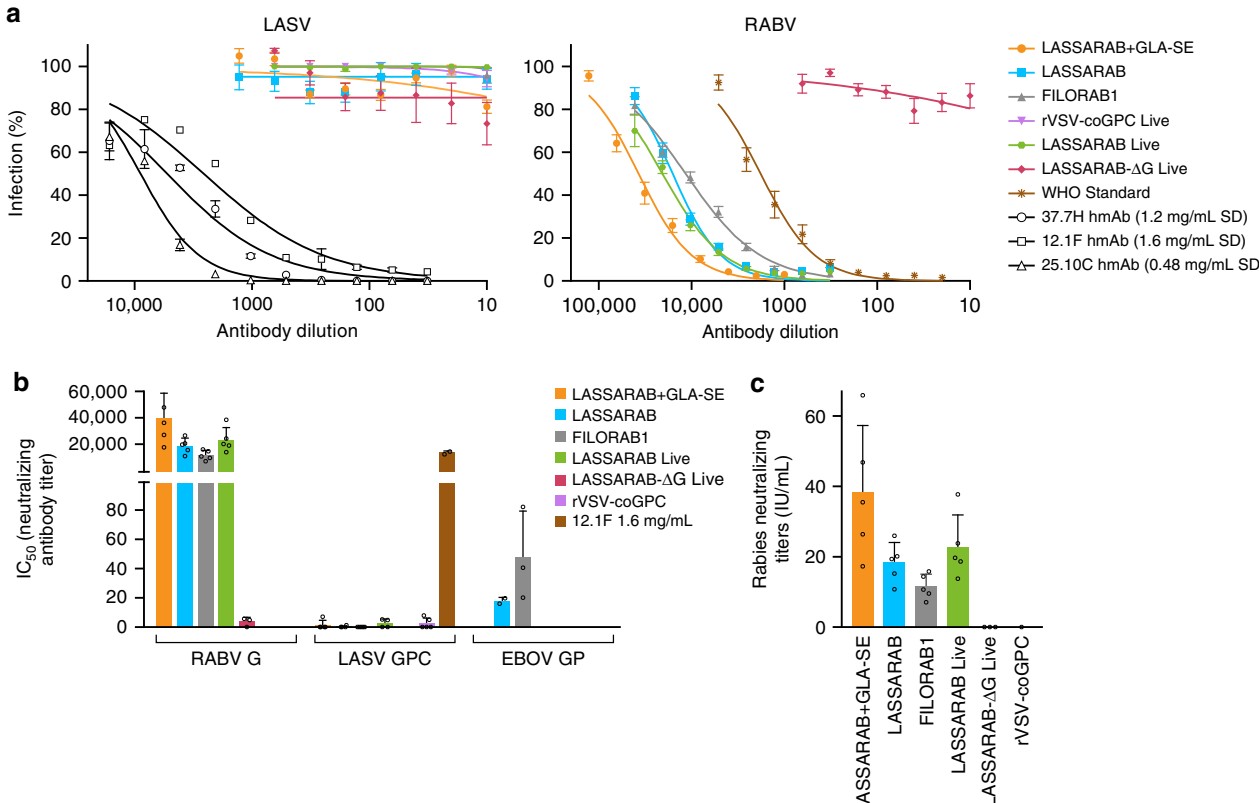

**Fig. 5** Virus neutralization antibody titers. Day 42 sera from immunized mice was incubated with pseudotyped rVSV-ΔG-NL-GFP. **a** rVSV-ΔG-NL-GFP was pseudotyped with either RABV-G, LASV-GPC, or EBOV-GP to assay for RABV-G, LASV-GPC, or EBOV-GP NAb titers, respectively. 12.1F, 37.7H, and 25.10c are LASV-GPC neutralizing antibodies used as a positive control for LASV-GPC neutralization[26]. Y-axis in **b** represents 50% of inhibitory serum dilution ($IC_{50}$) titers obtained based on the antibody dilution that has 50% infection percentage of infected cells curves obtained in **a**. All groups achieved high neutralizing titers against RABV-G except for the groups immunized with virus lacking RABV-G: rc-LASSARAB-ΔG and rVSV-coGPC, as expected. Regarding LASV-GPC pseudotyped VSVs, no immunization achieved appreciable amounts of neutralizing antibodies. Neutralization of LASV GPC pseudotyped viruses with 12.1F, 37.7H, and 25.10C had an average IC50 of 1546 ng/ml, 375 ng/ml, and 69 ng/ml, respectively. **c** Rabies neutralizing titers were calculated by using the $IC_{50}$ values of the WHO sera standard (2 IU/ml) serial diluted with rVSV-ΔG-NL-GFP pseudotyped with RABV G. WHO international units/ml (IU/ml) were then calculated using the following formula: (sample $IC_{50}$ titer)/(WHO standard $IC_{50}$ titer) × 2.0 (WHO IU/ml standard starting dilution). IU/ml from test sera is plotted Y-axis. All error bars represented are the SEM of triplicate values of 5 mice per group. (****$P < 0.0001$; ***$P < 0.001$; **$P < 0.01$; *$P < 0.05$)

(ADCC and ADCP) in vivo. Because non-NAb effector function in mice is dependent upon Fcγ receptor engagement, we used Fcγ chain KO mice (Fcγ$^{-/-}$)[50] to test whether non-NAb against LASV GPC are as relevant in protection against LF as our previous results suggest. To that end, we developed a surrogate LASV murine model utilizing rVSV-GPC (Supplementary Fig. 4), since LASV is a BSL-4 agent with no established LASV murine model. Because rVSV-GPC expresses LASV GPC as its sole glycoprotein, it should have a similar tropism to LASV, and such approach has been a strategy used elsewhere for other VHF viruses[51–53]. Mice were made more susceptible to rVSV-GPC by blocking the interferon-α/β receptor (IFNAR) with anti-IFNAR mAb followed by an IP exposure of rVSV-GPC 24 h later[54].

BALB/c (WT) and BALB/c Fcγ$^{-/-}$ mice were immunized twice with either LASSARAB+GLA-SE or FILORAB1+GLA-SE (controls) in a total of four groups (Fig. 8a). On day 42 post-primary immunization, mice were exposed IP with $10^4$ pfu of rVSV-GPC and clinical signs and weight were monitored (Fig. 8b and Supplementary Fig. 4). WT LASSARAB immunized mice mostly resisted infection, with 8/10 mice having only transient weight loss (Fig. 8b and Supplementary Fig. 4, continuous orange line). Meanwhile, all (10/10) of the Fcγ$^{-/-}$ LASSARAB mice quickly lost weight and succumbed to infection by day 5, with some showing signs of hemorrhage (Fig. 8b and Supplementary

Fig. 4 dashed orange lines) indicating that Fcγ is essential to control viral infection in LASSARAB immunize mice. In FILORAB1 immunized mice (control), both WT and Fcγ$^{-/-}$ groups had a similar outcome, with 2/5 mice of each group surviving infection until study endpoint (Fig. 8b and Supplementary Fig. 4, gray lines), demonstrating that both WT and Fcγ$^{-/-}$ are equally susceptible to surrogate LASV exposure.

Upon pre-exposure analysis of GPC-specific IgG titers, both WT and Fcγ$^{-/-}$ mice immunized with LASSARAB had significantly higher titers in comparison with FILORAB1 control mice (Fig. 8c and Supplementary Fig. 4c), but no LASV NAbs were detected in neutralization assays (Supplementary Fig. 4b). In post-exposure analysis of LASV NAbs, surviving LASSARAB immunized mice developed little to no neutralizing antibody (Fig. 8d, orange symbols), while one WT FILORAB1 vaccinated mouse developed modest levels of LASV NAbs (Fig. 8d). Overall this data shows that previous LASSARAB immunization is heavily dependent on non-NAb effector function activity in vivo for protection against LASV.

## Discussion

The WHO R&D Blueprint for Action to Prevent Epidemics[55] defines LF as a priority agent for vaccine development.

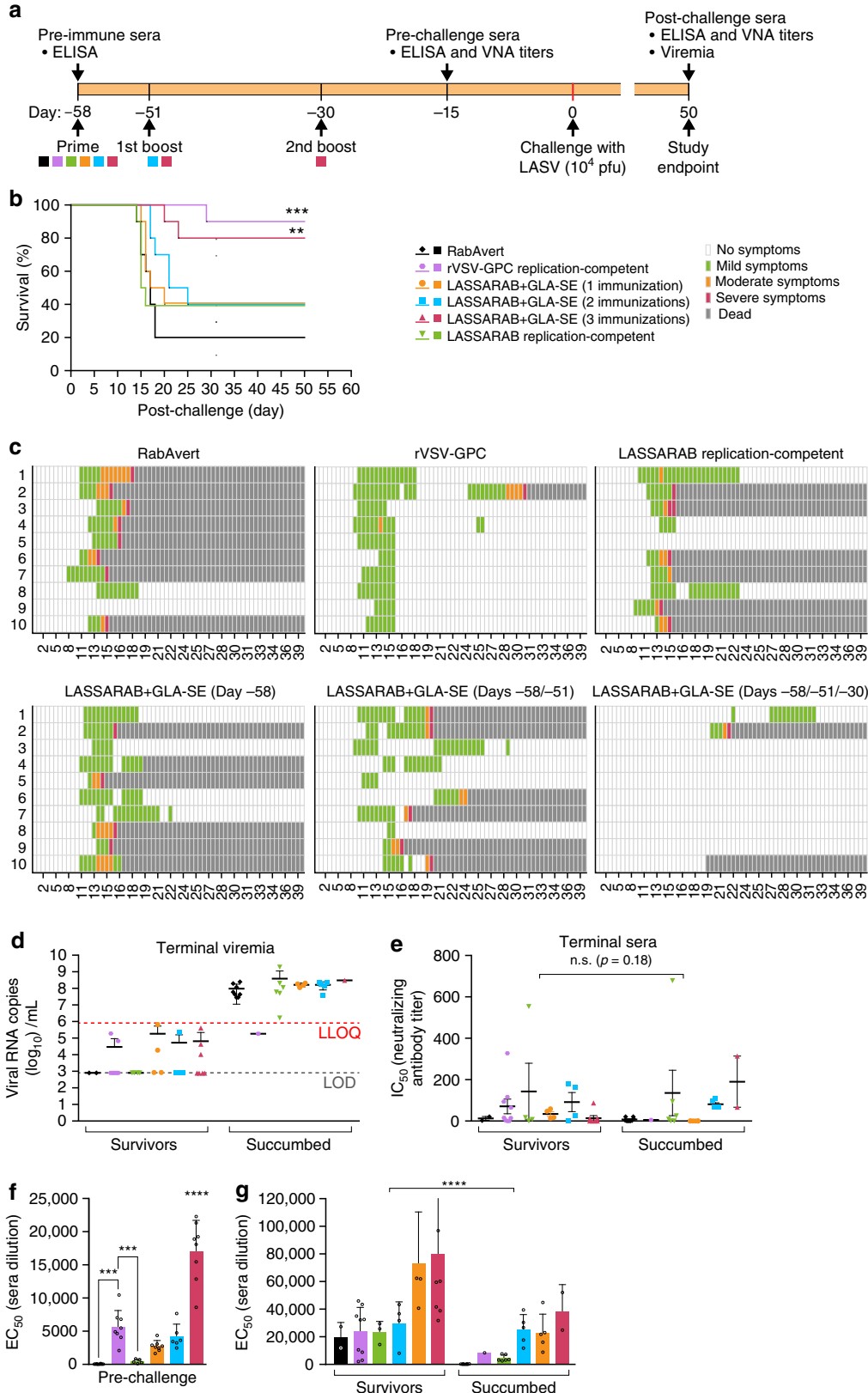

Accordingly, preferred vaccine requirements include: (1) a highly favorable risk-benefit profile suitable for all age groups, (2) practicality for non-emergency/preventive scenarios, (3) at least 90% efficacy in preventing disease, (4) high thermostability, and (5) the possibility of co-administration with other vaccines. LASSARAB appears to be the first inactivated LF vaccine to fulfill

most of these requirements as demonstrated in our study and based on previous work done with the same platform for other VHFs[37]. Another advantage to LASSARAB, as an inactivated LF vaccine, is that it could potentially be used in pregnant women and immunosuppressed patients, both of which are major risk groups for LF. In addition to LF, LASSARAB also confers

**Fig. 6** LASV challenge of outbred Hartley guinea pigs immunized with several vaccine candidates and control. **a** Guinea pigs were immunized with either two replication competent vaccines: rVSV-GPC (positive control for survival) and LASSARAB replication-competent at $10^6$ ffu by intraperitoneal injection (IP); or inactivated LASSARAB+GLA-SE with different immunization schedules: day −58 (LASSARAB+GLA-SE (1)), day −58, day −51 (LASSARAB+GLA-SE (2)) and day −58, day −51, and day −30 (LASSARAB+GLA-SE (3)). RabAvert was used as mock immunization (negative control). **b** Survival curves post IP exposure with $10^4$ pfu guinea pig-adapted LASV Josiah strain. Statistical significance is compared against Rabvert group using log-rank (Mantel–Cox) test. **c** Heat plot representing the clinical score information. X-axis represents days' post-challenge and Y-axis represents the individual animal number. **d** Terminal viremia was plotted using LASV RNA copies/ml in Y-axis. Statistical significance was calculated using Kruskal–Wallis one-way ANOVA (not significant). **e** LASV neutralizing antibody titers is reported as the $IC_{50}$ (half maximal inhibitory concentration) of serum dilution. The human mAbs 25.10C, 12.1F, and 37.7H[25,26] were used as positive LASV neutralization controls. **f** Pre-challenge titers of LASV GPC specific IgG were performed on sera collected on day −15 prior to challenge by ELISA with LASV GPC antigen and the $EC_{50}$ (50% effective concentration) of serum dilution was plotted in the Y-axis. Statistical significance (compared to the RabAvert group) was calculated by using one-way ANOVA (post-hoc test Tukey Honest Significant Difference Test). **g** Post-challenge titers of LASV GPC-specific IgG was performed on sera collected on terminal bleeding of both succumbed animals and survivors (day 50 post challenge) and the $EC_{50}$ of serum dilution is plotted on the Y-axis. Statistical significance reported between survivors and succumbed in **e**, **g** was determined by using two-way ANOVA. All error bars represented are the standard error mean (SEM) of 10 animals per group (in triplicates). (****$P < 0.0001$; ***$P < 0.001$; **$P < 0.01$; *$P < 0.05$)

protection to rabies (Fig. 5b), which is a major health burden in Africa[56].

Most LASV vaccine studies have characterized the role of humoral response against LASV as either a secondary mechanism of protection or even detrimental to survival[11]. Such correlations were drawn based on results measuring antibody responses against LASV nucleoprotein (NP) or nonspecific LASV antigens[11,16,57]. Although NP is highly immunogenic, it is neither expressed on the surface of cells nor virions. As such, antibodies directed against LASV NP should only have diagnostic value. Meanwhile, GPC has been shown to be the most effective LASV immunogen but, to our knowledge, no attempts were made to correlate GPC-specific humoral response with LASV protection[16,58–60]. Thus, as part of LASSARAB characterization, we were compelled to develop a GPC-specific antigen that is expressed in its native conformation (Supplementary Fig. 1). Throughout the development of LASSARAB, we observed that replication-competent LASSARAB and replication-competent LASSARABΔG were poor inducers of GPC-specific antibodies, despite being able to induce RABV protective response (Figs. 4c, 5, and Supplementary Fig. 2). In contrast, when inactivated LASSARAB immunizations were combined with a late boost (day 28 post-prime), high levels of LASV GPC-specific antibodies were induced at later time points, especially when administered with a TLR-4 agonist (GLA-SE). This contrast might be attributed to the fact that LASV GPC is a poor immunogen[28,61] and, as such, induction of antibodies against GPC might be dependent on replication-competent vectors that achieve high or persistent viral loads post immunization. Given that inactivated LASSARAB incorporates LASV GPC, it can safely be administered in higher dosages in a prime/boost regimen and, as such, more antigen might be available to prime follicular B helper T cells and B cell response. The high effectivity of a TLR-4 agonist in inducing higher levels of anti-LASV GPC antibodies with higher quality (IgG2c bias) further corroborates recent findings by Galan-Navarro et al.[61] indicating that inactivated LASV vaccines might benefit of TLR-4 agonists. Nonetheless, no NAbs against LASV pseudotypes were detected in either replication competent or inactivated approaches (Figs. 4, 5 and Supplementary Fig. 3 and 4).

Because it has been the case with vaccines for some other viruses[44,62,63], it might be expected that an effective LF vaccine protects through NAbs. Sommerstein et al. have elegantly demonstrated that LASV exposure or immunization in mice does not induce LASV NAbs due to the LASV GPC's glycan shield[28]. Additionally, as recently shown by the important works of Robinson JE et al. and Hastie et al., most potent LASV NAbs (such as 37.7H) require very specific quaternary epitopes bridging

LASV GP1 and GP2, making it challenging to elicit through immunization. Interestingly, these NAbs, instead of blocking GPC receptor binding, achieve neutralization by stabilizing LASV GPC in its pre-fusion conformation[25,26,42]. The lack of NAbs induced with the several vaccine candidates, either replication competent or inactivated, in our study (Fig. 5a) and in previous published vaccine candidates, further corroborates this expectation[11,23]. Even after LASV exposure, only a small fraction of human and animal survivors produce NAbs, findings that our study further confirmed (Figs. 6, 8)[11,26,28]. Additionally, we showed that guinea pigs that succumbed to disease also had NAbs, suggesting either that NAbs by themselves play a minor role in protection or that they develop too late during infection to impact outcome. Studies by Mire et al. have recently shown that some LASV NAbs can mediate protection in NHPs and guinea pigs when administrated prophylactically[25,26]. Although providing evidence that GPC specific mAbs can mediate protection against Lassa Fever, the role of antibody-dependent effector cellular functions was not evaluated and GPC-specific non-NAbs were not used. Furthermore, LASV neutralizing potency in vitro did not necessarily correlate with protection[27]. Together with the findings in our study (Fig. 6), this raises the question whether GPC-specific non-NAb play a role in protection through other mechanisms, such as ADCC, since guinea pig survival post-LASV exposure was correlated with high levels of GPC-specific non-NAb independent of NAb titer.

In several other viruses (e.g., Influenza, LCMV), antibody Fc-FcγR interactions leading to ADCC and ADCP are important for protection, playing a critical role both in viral clearance and in preventing chronic infection regardless of neutralizing ability[29,30,32,34,64]. Through our in vitro studies, we showed that sera from LASSARAB-immunized mice with high GPC-specific antibodies (Supplementary Fig. 4) did not neutralize LASV but elicited significant ADCC and ADCP of 3T3 cells expressing LASV GPC (Figs. 5, 7). Interestingly, the Fcγ-RIV blockade did not reduce ADCP activity by macrophages (Fig. 7e), despite having a high affinity for IgG2 subclass-dependent ADCP. This suggests that GPC-specific IgG1 might be mediating ADCP[65]; nevertheless, in contrast with IgG2 subclass, GPC-specific IgG1 titers were almost non-existent in the purified IgG used (S3c).

To corroborate the relevance of Fcγ-R effector functions in LASSARAB-induced protection in vivo, we used an Fcγ-KO mouse model challenged with surrogate LASV exposure (Fig. 8)[50]. This approach permitted us to dissect the role that LASSARAB induced non-NAb play in protection against surrogate LASV exposure in the context of a similar immunogenic response. Despite similar levels of antibody titers and isotype to both RABV G and LASV GPC as detected by ELISA (Fig. 8c and

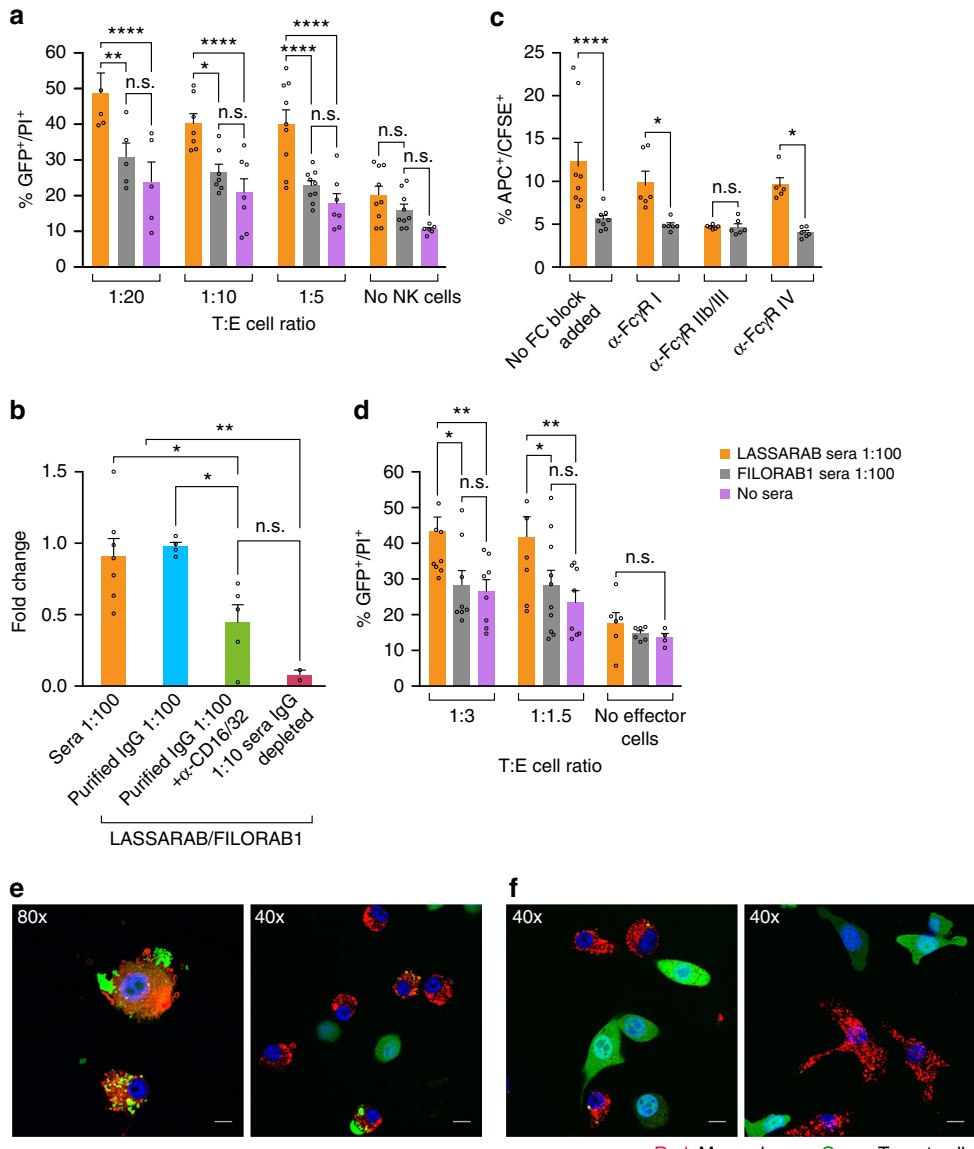

**Fig. 7** Evaluation of antibody effector cell functions mediated by murine NK and macrophage cells against 3T3 expressing LASV GPC. Day 42 sera from immunized mice was incubated with 3T3-LASV cells and 30 min either murine NK or macrophage cells were added and results were analyzed 4 h later by either flow cytometry (Supplementary Fig. 3 and **a–d**) or confocal microscopy (**e**). Purified murine C57BL/6 NK cells (**a**) or IC-21 macrophages (**d**) were added at different Target:Effector cell ratios (T:E) with target cells incubated with either LASSARAB sera (yellow), FILORAB1 sera (gray) or no sera (pink). The Y-axis represents the percentage of cellular cytotoxicity based on GFP+/PI+ cells (gating strategy and flow plots in Supplementary Fig. 3). **b** To determine which antibody isotype class is important for ADCC, NK cells were added at 1:5 T:E and incubated with either unprocessed sera (sera 1:100 condition), purified IgG (20 µg/ml), or IgG impoverished sera (1:5 dilution) from LASSARAB and FILORAB1 immunized mice. The Y axis represents cytotoxicity fold change of LASSARAB sera or IgG compared to FILORAB1 sera or IgG with same respective conditions. Anti-CD16/32 (FcγR-II/III) was also added at 25 µg/ml to confirm that FcγR blockade reduces ADCC activity. **c**, **e**, **f** To analyze ADCP J774.A1 macrophages were added at 1:5T:E or 1:1T:E (confocal) to 3T3-LASV cells incubated with either LASSARAB sera (**c**, **e**) or FILORAB1 sera (**c**, **f**). In **c** anti-CD16.2 (Fcγ-RIV), anti-CD16/32 (Fcγ-RII/III), and anti-CD64 (Fcγ-RI) were added at 25 µg/ml to check the effect of different FcγR blockade on ADCP activity. All error bars are the SEM of at least three independent experiments executed with duplicates. All statistical significance represented was performed through either a one- or two-way ANOVA and using a post-hoc analysis Tukey Honest Significant Difference test. (****$P < 0.0001$; ***$P < 0.001$; **$P < 0.01$; *$P < 0.05$). The bar in **e** indicates 6 µm (80×) or 12 µm (40×)

Supplementary Fig. 5a and b), LASSARAB immunized Fcγ-KO mice quickly succumbed to surrogate LASV exposure, in contrast to the WT mice. However, some differences exist between human and mouse Fcγ-Rs, and future studies using humanized knock-in models would be of interest[66]. Curiously, besides the critical role that Fcγ-R effector functions played in protection against LASV, our results from Fig. 8 indicated (but not significantly) that Fcγ$^{-/-}$ mice immunized with LASSARAB seemed more susceptible

to surrogate LASV infection than control mice (Fig. 8 and Supplementary Fig. 5). Although based on a contrived model, this makes us question whether, beyond viral clearance, pre-existing GPC-directed non-NAbs might also work as immune regulators in LASV infection.

By the end of our guinea pig exposure study (Fig. 6d), we observed that ~ 20% to ~ 40% of survivors had low (below the LOQ) but detectable levels of LASV RNA in the blood 50 days'

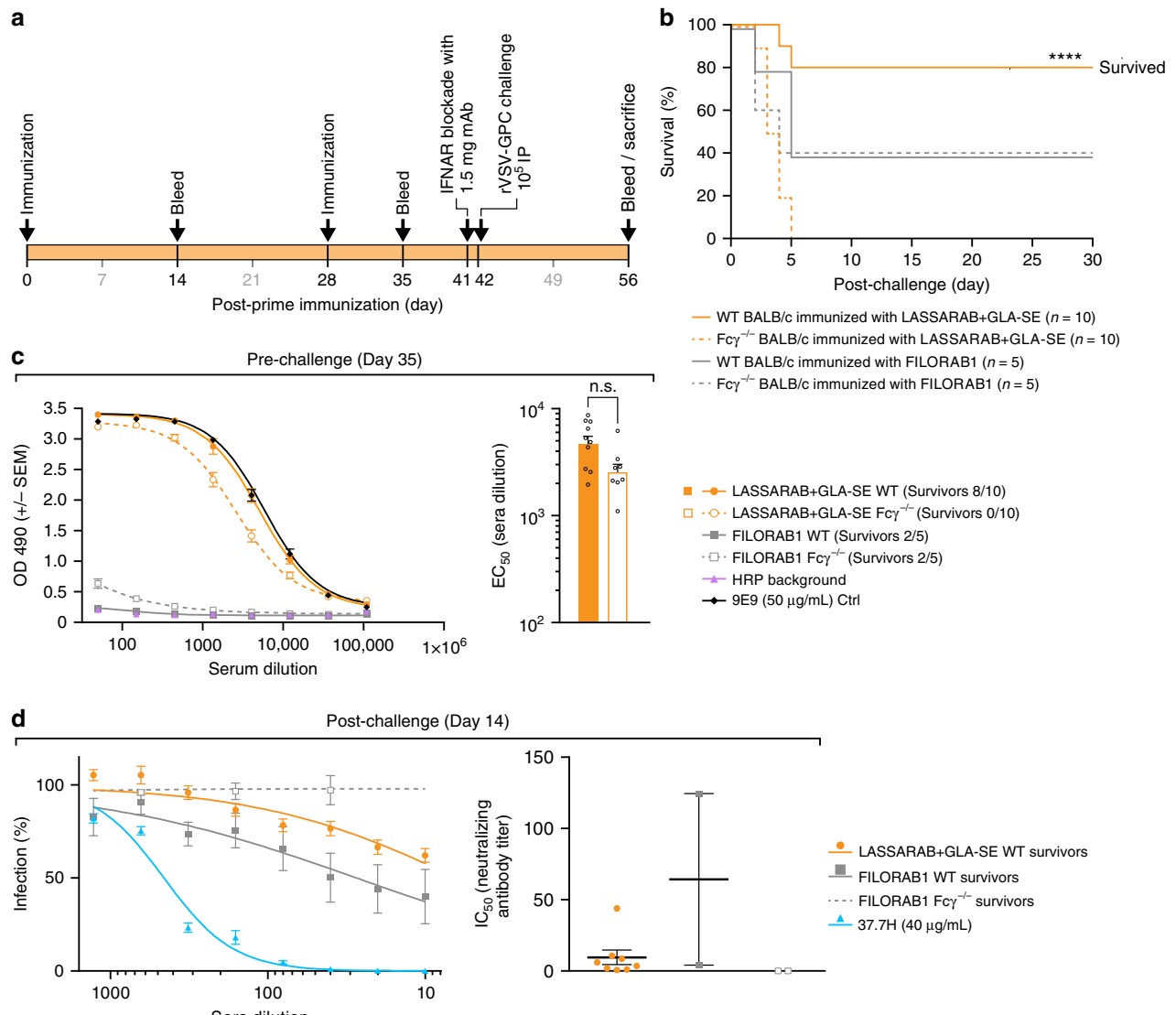

**Fig. 8** Evaluation of in vivo relevance of non-NAbs LASV GPC specific antibodies induced by LASSARAB + GLA-SE vaccination. **a** 8- to 10-week-old Balb/c (WT) or Balb/c with Fcγ chain KO (Fcγ$^{-/-}$) female mice were immunized with 10 μg of inactivated particles of either LASSARAB or FILORAB1 (mock control) on day 0 and boosted on day 28. All four groups in total were adjuvanted with 5 μg of GLA in a 2% SE with each vaccination. One day before exposure (day 41) animals were injected with 1.25 mg of anti-Ifnar mAb (MAR1-5A3, Leinco technologies) through intra-peritoneal injection (IP). On day 42, mice were exposed to 10$^4$ rVSV-GPC virus IP and general health (weights and clinical observation) was recorded until endpoint criteria were reached or end of study (supplemental). **b** Survival curves post-exposure of rVSV-GPC. Significance is compared between the WT LASSARAB vaccinated and the Fcγ$^{-/-}$ vaccinated using the log-rank (Mantel–Cox) test. **c** Pre-exposure total IgG titers anti LASV GPC were measured by ELISA on day 35 post-prime and ELISA curves were plotted according to OD490 reading value (Y-axis) and serum dilution (X-axis). On the right, EC$_{50}$ (half maximal effective concentration) of serum dilution of both LASSARAB groups (WT and Fcγ$^{-/-}$) is plotted on Y-axis on a log scale; statistical significance was calculated using one-way ANOVA. **d** Virus neutralization assay using pseudotyped VSV-GFP-NanoLuc with LASV GPC. On the right, percentage of cells infected is plotted against the serum dilution (survivors on day 14 post-exposure) of each respective group. On the right, the IC$_{50}$ (half maximal inhibitory concentration) of serum dilution is plotted individually and significance was calculated using one-way ANOVA. Error bars represent Standard Error Mean (SEM) and include all mice (n = 10 per group [WT and KO] in LASSARAB and n = 5 per group [WT and KO] in FILORAB1 control) in pre-challenge and survivor mice in post-challenge. (****P < 0.0001; ***P < 0.001; **P < 0.01; *P < 0.05)

post-exposure in all groups, except LASSARAB and RabAvert. This has been reported in the literature for LASV in NHPs[23,25]. This result suggests a chronic asymptomatic infection that, after reactivation, may explain some of the late deaths and clinical signs observed in both the LASSARAB+GLA-SE and rVSV-GPC groups. As such, future studies should consider possible LASV chronicity and reactivation.

As a major LF surge unfolds in Nigeria at the time of manuscript preparation, the necessity to fully understand the immunomechanisms of protection of LASV becomes an increasingly

important and crucial task for LF vaccine development. Ideally, a LF vaccine should be protective, safe, and confer a long-lasting humoral immunity that can be easily measured and identified as a correlate of protection. As our results demonstrate, LASSARAB induces high LASV GPC-specific IgG titers that correlate with protection prior to LASV exposure, in the absence of LASV NAbs. This could potentially become a LF correlate of protection that would provide easy screening for vaccine efficacy post immunization. Additionally, the finding that GPC-specific non-NAbs play a crucial role in protecting mice against a LASV

surrogate exposure suggests that non-NAb cellular effector functions should be further investigated as a correlate of protection in both LF vaccine development and mAb antibody therapy.

## Methods

**Generation and recovery of Rhabdovirus vaccine vectors**. To generate the vaccine vectors LASSARAB, LASSARABΔG, and rVSV-GPC, the ORF of LASV GPC Josiah strain was codon-optimized for mammalian codon-usage and synthetized by GenScript (Genbank, Accession Number MH778559). LASV GPC was cloned between BsiWI and NheI restriction digest sites of BNSP333, generating LASSARAB. LASSARABΔG was generated by removing the RABV glycoprotein (G) from the LASSARAB cDNA using the PacI and SmaI restriction digest sites and subsequent re-ligation after treatment with Klenow Fragment (Promega).

rVSV-GPC was generated by replacing the native VSV G, through MluI and NheI restriction digestion site, with a codon-optimized LASV GPC (above) amplified by the PCR primers RLP3 and RLP4 containing the MluI or NheI restriction sites and cloned in cVSV-XN vector[62]. The correct sequence of all the three plasmids were confirmed by sequencing using RP951, RP952, VP5, and VP6 primers.

Recombinant RABV and VSV vaccines were recovered as described previously[67,68]. Briefly, X-tremeGENE 9 (Sigma-Aldrich®) in Opti-MEM (Gibco®) was used to co-transfect the respective full-length viral cDNA clones along with the plasmids encoding RABV N, P, G, L or VSV N, P, L proteins, and pCAGGs plasmids expressing T7 RNA polymerase in Vero cells in 6-well plates (RABV), or 293T cells in T25 flasks (VSV). The supernatants of RABV transfected cells were harvested after 7 days and after 3 days for VSV. Presence of infectious virus was detected by immunostaining for RABV N with 1:200 dilution of FITC anti-rabies monoclonal globulin (Fujirebio®, product # 800-092) or for virus-induced cytopathic effect (CPE) in the case of VSV.

**Request for material**. Upon reasonable request all utilized antibodies, plasmids, and viruses are available from the authors pending on an executed MTA as well as biosafety approval of the requesting institution(s).

**Cell culture**. Vero (ATCC® CCL81™), 293T (ATCC® CRL-3216™), and BSR (available from our laboratory) cells were cultured using DMEM (Corning®) with 5% FBS (Atlanta-Biologicals®) and 1% P/S (Gibco®)[36]. J774.A1 (ATCC® TIB-67™) macrophages, NIH/3T3 (ATCC® CRL-1658™), and their stable cell line derivatives were cultured using DMEM with 10% FBS and 1% P/S. IC-21 (ATCC® TIB-186™) macrophages were cultured using RPMI (Corning®) with 10% FBS and 1% P/S.

**Antibodies**. Mouse monoclonal antibodies (mAb) anti-LASV GPC (4C8, 9E9, and 5A3) were produced and provided by Dr. Gene Tan (J. Craig Venter Institute, La Jolla, CA). The human mAbs anti-LASV GPC (3.3B, 22.5D, 37.7H, 25.10C, and 12.1F) were a generous gift from Dr. Robert Garry (Tulane University)[26]. Rabbit polyclonal antibody (pAb) anti-LASV GPC was generous gift from Dr. Stephan Guenther (Bernhard-Nocht-Institute for Tropical Medicine, Hamburg, Germany). 4C12 human anti-RABV G mAb was a generous gift from Dr. Scott Dessain (Lankenau Institute for Medical Research, Wynnewood, PA)[37,62,63].

**Viral production and tittering**. LASSARAB, LASSARAB-ΔG, FILORAB1, rVSV-GPC, and SPBN viruses were grown and titered on Vero cells. For virus production, Vero cells were cultured with Opti-Pro serum-free media supplemented with 1% P/S and 4 mM L-Glutamine (Gibco®) and inoculated with a multiplicity of infection (MOI) of up to 0.01 of each respective virus. Viruses were harvested up to a total of six times with media replacement (Opti-Pro) or until 80% cytopathic effect was detected. Tittering was performed by limiting dilution focus-forming assay using RABV N with 1:200 dilution of FITC anti-rabies monoclonal globulin (Fujirebio®; catalogue number: 800-092). rVSV-GPC titers were determined by plaque forming assay using 2% methyl cellulose overlay.

**Purification and virus inactivation**. To produce inactivated LASSARAB and FILORAB1[38] (kind gift of Drishya Kurup, Thomas Jefferson University) vaccines, viral supernatant were sucrose purified and inactivated[37]. Briefly, viral supernatants were concentrated at least 10x by Amicon® stirred cell concentrator using a 500 kDa exclusion PES membrane (Millipore®) and centrifuged at $110,000 \times g$ through a 20% sucrose cushion. Virion pellets were resuspended in $1 \times$ DPBS (Corning®) containing 2% sucrose and betapropiolactone (BPL) (Sigma-Aldrich®) was added at a 1:2000 dilution for inactivation. Samples were left at 4 °C O/N shaking and next day BPL was hydrolized at 37 °C for 30 min.

**Adjuvant formulations**. The Toll-like receptor 4 agonist glucopyranosyl lipid adjuvant-stable emulsion adjuvant (GLA-SE) was produced by IDRI[38,69]. Formulation with inactivated vaccines was conducted prior to injection with a total 5 μg of GLA for mice or 7.5 μg of GLA for guinea pigs in a final v/v 2% SE concentration.

**Immunofluorescence**. Vero cells were seeded on glass coverslips and infected at an MOI of 0.1 with the respective viruses. 48 h later (24 for VSV constructs), cells were fixed with 2% paraformaldehyde (PFA) and probed with 10 μg/ml anti-RABV G mAb (4C12) and mouse 50 μg/ml of anti-LASV GP2 mAb (9E9). Secondary goat polyclonal antibody (Jackson ImmunoResearch® catalogue numbers: 109-225-088; 115-165-146) anti-human IgG and anti-mouse IgG conjugated with Cy2 and Cy3 dyes, respectively, were used at 4 μg/ml. Slides were mounted with DAPI containing mounting media (VECTASHIELD®) and images were taken with a Zeiss AxioSkop 40 microscope and color channels were compiled using ImageJ software (OSS NIH).

**Viruses and ELISA antigen characterization**. Virus particles and purified LASV GPC were denatured with Urea Sample Buffer (125 mM Tris-HCl [pH 6.8], 8 M urea, 4% sodium dodecyl sulfate, 50 mM dithiothreitol, 0.02% bromophenol blue) at 95 °C for 5 min. 2 μg of protein was resolved on a 10% SDS–polyacrylamide gel and stained O/N with SYPRO Ruby (Thermofisher) for total protein analysis. For western blot analysis SDS-PAGE gel was transferred onto a nitrocellulose membrane in Towbin buffer (192 mM glycine, 25 mM Tris, 20% methanol) then blocked in 5% milk dissolved in PBS-T (0.05% Tween 20) at room temperature for 1 h. Next, the membrane was incubated O/N with either rabbit pAb anti-LASV GPC or 9E9 mAb anti-LASV GP2 at a dilution of 1:1000 in 5% bovine serum albumin (BSA). Rabies G and P proteins were confirmed with a rabbit anti-G and P polyclonal antibody used at 1:1000[62]. After washing, the blot was incubated for 1 h with horseradish peroxidase (HRP)-conjugated anti-rabbit or mouse IgG diluted (Jackson ImmunoResearch® catalogue numbers: 115-035-146; 111-035-144) at 1:50,000 in 1% milk PBS-T. Proteins were detected with SuperSignal West Dura Chemiluminescent substrate (Pierce®).

**Animals ethics statement**. Mice and guinea pigs used in this study were handled in adherence to both the recommendations described in the Guide for the Care and Use of Laboratory Animals, and the guidelines of the National Institutes of Health and the Office of Animal Welfare. Animal work was approved by the Institutional Animal Care and Use Committee (IACUC) of Thomas Jefferson University (TJU) or the National Institutes of Health, National Institute of Allergy and Infectious Diseases, Division of Clinical Research Animal Care and Use Committee for experiments performed at each respective facility. Animal procedures at TJU were conducted under 3% isoflurane/O₂ gas anesthesia. Mice were housed with up to five individuals per cage, under controlled conditions of humidity, temperature, and light (12-h light/12-h dark cycles). Food and water were available ad libitum.

**Viral pathogenicity evaluation**. Five groups of five 6- to 8-week-old female Swiss Webster mice were either intranasally (IN) or intraperitoneally (IP) infected with $10^5$ PFU/FFU of each of the respective viruses diluted in 20 μl phosphate-buffered saline (PBS). Mice were weighed daily and monitored for signs of disease until day 28 post-infection. Mice that lost more than 20% weight or showed severe neurological symptoms were humanely euthanized. Intracranial challenge (IC) was performed in 48, 6- to 8-week-old Balb/c mice were anesthetized using isoflurane to effect, followed by IC injection of 10 fold increasing dose from $10^2$ to $10^5$ FFU of infectious virus. Mice were monitored daily for up to 21 days post-exposure. Mice were euthanized when signs of neurological disease, including tremors, seizure, prostration, and paralysis, were observed using a pre-determined scale of severity. Forty-eight, 3- to 4-day-old Swiss Webster mice were anesthetized by hypothermia followed by IC injection of 10-fold increasing dose of virus from $10^2$ to $10^5$ ffu of infectious virus. Mice were monitored daily for signs of neurological disease and euthanized when signs developed or at 10 days post-exposure.

**Humoral immunogenicity evaluation in mouse model**. Five groups of five 6- to 8-week-old female C57BL/6 mice were immunized intramuscularly (IM) with $10^6$ PFU/FFU of live virus diluted in PBS or with 10 μg BPL-inactivated virus (3 doses at 0, 7, and 28 days) formulated in either PBS or GLA-SE adjuvant (see Fig. 4 and adjuvant formulation below). All IM immunizations were performed by administering 50 μl of live or BPL-inactivated virus into each hind leg muscle. For serum collection, retro-orbital bleeds were performed under isoflurane anesthesia on days 0, 7, 14, 21, 28, and 35, with the final bleed on day 42 or 63.

**LASV challenge on outbred Hartley guinea pigs**. Six groups of ten Hartley guinea pigs with PinPorts for blood withdrawal (Charles River Laboratory) were vaccinated as follows: Group 1: Mock (PBS), Group 2 rVSV-GPC $10^7$ FFU, Group 3 RABV-LASV-GPC $10^7$ FFU, Group 4 RABV-LASV-GPC (30 μg) + GLA-SE (7.5 μg) on day −58 of virus exposure, Group 5 RABV-LASV-GPC (30 μg) + GLA-SE (7.5 μg) on days −58 and −51 of virus exposure, Group 6 RABV-LASV-GPC (30 μg) + GLA-SE (7.5 μg) on days −58, −51 and −30 of virus exposure. All subjects were challenged with 10,000 PFU of guinea pig-adapted LASV (GPa-LASV) (IRF0205; L segment GenBank KY425651.1; S segment GenBank KY425650.1) by IP route[48]. Subjects were monitored at least once daily throughout the experiment and at least twice daily following virus exposure until clinical signs of disease abated. Blood withdrawals were performed at days −65, −58, −51, −30, 0, 16 and study end at day 42 post-exposure. All LASV experiments were performed in a biosafety level 4 environment and subjects were anesthetized using isoflurane/O₂ gas

anesthesia for all procedures. Clinical scoring to determine euthanasia was based on the appearance of one of the following clinical changes: change in skin and mucous membrane color, unthrifty appearance, unresponsiveness, agonal breathing, paralysis, head tilt, persistent scratching, tremors. Subjects that met endpoint criteria and subjects that survived to study end, day 42, were humanely euthanized and a complete necropsy was performed.

**In vitro ADCC/ADCP evaluation**. The sera used for these assays were collected on day 42 from two groups of five mice each IM immunized with 10 μg BPL-inactivated LASSARAB or FILORAB1 (two doses: at day 0 and at day 28) formulated with GLA-SE adjuvant. Serum collected from individual mice was pooled and heat inactivated for 30 min at 56 °C. For IgG purification, serum was run through a protein G high performance Spintrap column (GE Healthcare).

**Surrogate LASV challenge on mouse model**. Four groups of either Balb/C or Fcγ knockout Balb/C (Balb/C Fcγ$^{-/-}$ generously donated by Dr. Jeffrey V. Ravetch, Rockefeller University) were IM immunized with 10 μg BPL-inactivated LAS-SARAB or FILORAB1 (two doses: at day 0 and at day 28) formulated with GLA-SE adjuvant and sera were collected on day 0 or day 35 post-immunization. On day 41, mice were injected IP with 1.25 mg of mouse anti-IFAR1 mAb clone: MAR1-5A3 (Leinco Technologies, catalogue number: I-401). On day 42, mice were injected with 10$^4$ pfu of rVSV-GPC diluted in PBS. rVSV-GPC was previously confirmed to be pathogenic in immunosuppressed mice by titering the virus to the least amount that causes 100% lethality on naïve Balb/C mice. Health and weight were monitored daily. Mice were sacrificed when: (1) weight loss reached >20% or (2) if severe clinical signs of disease were observed. Terminal bleeding was collected upon sacrifice when possible.

**Enzyme-linked immunosorbent assay (ELISA)**. Individual mouse or guinea pig serum was analyzed by ELISA for the presence of IgG specific to LASV GPC, RABV G, and EBOV GP. Antigens were resuspended in coating buffer (50 mM Na$_2$CO$_3$ [pH 9.6]) at a concentration of 500 ng/ml and then plated in 96-well immulon 4 HBX plates (Nunc®) at 100 μl in each well and incubated O/N for 4 °C. Plates were then washed three times with PBS-T (0.05% Tween 20 in 1×PBS), blocked for 1 h (5% milk in PBS-T), washed three times with PBS-T, and then incubated O/N at 4 °C with three-fold serial dilutions of sera or control mAb (starting at either 1:50 or 1:150 dilution) in PBS containing 0.5% BSA. Next, plates were washed three times, followed by the addition of either horseradish peroxidase (HRP) conjugated goat anti-mouse: IgG (H+L), Fc specific (heavy chain), IgG2c, IgG2a, and IgG1; or goat anti-guinea pig Fc-specific (heavy chain) secondary antibody at 1:10,000 dilution in PBS-T (Jackson ImmunoResearch® catalogue numbers: 115-035-146; 115-035-071; 115-035-205; 115-035-206; 115-035-208; 106-035-008). After incubation for 2 h at RT, plates were washed three times with PBS-T, and 200 μl of o-phenylenediamine dihydrochloride (OPD) substrate (Sigma-Aldrich) was added and left incubating for exactly 15 min. The reaction was stopped by adding 50 μl of 3 M H2SO4. Optical density was determined at 490 nm (OD490). ELISA data was analyzed with GraphPad Prism 7 using a sigmoidal nonlinear fit (4PL regression curve) model to determine the half maximal Effective Concentration (EC50) serum or antibody titer.

**Generation and production of ELISA antigens**. RABV G antigen was generated as described[36]. Briefly, RABV G and LASV GPC antigen were generated by infecting BSR cells with either rVSV-GPC (for LASV GPC antigen) or SPBN (RABV G antigen) in Opti-Pro SFM (Gibco®). Viral supernatants were concentrated and purified as described in virus purification methods section (see above). Viral pellets were then resuspended in TEN buffer (100 mM NaCl, 100 mM Tris, 10 mM EDTA pH7.6) containing 2% OGP (Octyl β-D-glucopyranoside) detergent and incubated for 30 min while shaking at RT. Mixture was then centrifuged at 3000 g, and the supernatant was collected and further centrifuged at 250000 g for 90 min. Supernatant was collected and analyzed by SDS-PAGE and WB for LASV GP1 and GP2 presence (see above).

**Virus neutralization assay**. Virus neutralization assay (VNA) was conducted based on a modified VSV based VNA[38], by generating a single round VSV pseudotype reporter virus (ppVSV-NL-GFP) expressing nano-luciferase (Nano-Luc® Promega) and GFP.

**Generation of VSV pseudovirons (ppVSV)**. To generate ppVSV, the cDNA plasmid backbone of rVSV-GPC was digested with MluI and NheI restriction enzymes to remove the LASV GPC glycoprotein and insert the NanoLuc ORF (Promega). To enable GPC expression, the EGFP ORF plus a VSV start stop signal were inserted in XhoI and NheI cloning sites. Viruses were recovered as described above and further propagated on BSR cell line expressing VSV-G. To pseudotype ppVSV-NL-GFP with either LASV GPC, RABV G, or EBOV GP, 293T cells were transfected with pCAGGS plasmid encoding either LASV GPC (Josiah strain), RABV G (SAD-B19 strain), or EBOV GP (Mayinga strain), respectively, using X-tremeGENE 9 (Sigma-Aldrich) as a transfection reagent. 24 h post transfection,

pVSV-NL-GFP was added to the cells at an MOI of 1 and viral supernatant was collected 24 and 48 h later.

**Virus neutralization assay (VNA)**. For VNA using animal sera (mouse or guinea pig), the serum was heat inactivated at 56 °C for 30 min to ensure complement deactivation. Next, heat-inactivated serum was diluted two fold starting at 1:10 dilution (1:100 in RABV-G pseudotyped assays) in Opti-MEM (Gibco), and 10$^4$ ppVSV-NL-GFP particles were added to each dilution series. Control mAbs (12.1F, 25.10C, 37.7H, and 9E9, see Antibodies section above) and WHO international standard sera were added starting at 30 μg/ml and 2 UI/ml, respectively. The sera/antibody+virus mix was incubated for 2 h at 34 °C with 5% CO$_2$ and transferred to a previously seeded monolayer of Vero cells in a 96 well plate and further incubated for 2 h at 34 °C with 5% CO$_2$. Next, the virus/serum mix was replaced by complete DMEM media. At 18–22 h later, cells were lysed with passive lysis buffer (Promega) and transferred to an opaque white 96-well plate, with NanoLuc® substrate (Promega) added following the manufacturer's recommendations. Relative luminescence units were normalized to 100% infectivity signal as measured by no sera control (maximum signal). Half maximal inhibition (IC$_{50}$) values were calculated by GraphPad® Prism 7 using a sigmoidal nonlinear fit model (4PL regression curve). Values that were above 100% infectivity were converted to 100%.

**RT-PCR analysis for LASV viral loads**. See also refs.[70]. 200 ul of whole blood was lysed for RNA extraction using Trizol LS at a 3:1 vol:vol ratio. RNA samples were then extracted using the QIAMP Viral RNA Mini Kit (QIAGEN) and eluted in 50 μl Buffer AVE (QIAGEN). 5 uL of extracted RNA per reaction was added to 2X Master Mix with Superscript III Platinum One Step qRT-PCR kit (Invitrogen) with final concentrations of 1 μM forward primer (5′CCACCATYTTRTGCATRTGC CA), 1 μM reverse primer (5′GCACATGTNTCHTAYAGYATGGAYCA) and 0.1 μM probe (FAM_AARTGGGGYCCDATGATGTGYCCWTT). Cycling conditions were 45 °C for 15 min for reverse transcription, 95 °C for 2 min, followed by PCR amplification for 45 cycles at 95 °C for 15 s, then 60 °C for 30 s on an ABI 7500 real-time PCR system (Applied Biosystem®). In-vitro transcribed RNA was used as the standard. The LASV sequence from 3255 to 3726 (Genbank accession number: KY425634.1) was cloned under a T7 promoter in vector pCMV6-AC. The fragment was linearized and 1ug of DNA was used in the in-vitro transcription reaction using the MEGAscript T7 transcription kit (Ambion). RNA copy number was calculated and 1:10 dilutions were made to provide a standard from 9log$_{10}$ viral RNA copies to 1log$_{10}$ viral RNA copies. Quantification was performed by CT analysis (Applied Biosystem®).

**Target cell generation for ADCC and ADCP**. Target cell generation (3T3-LASV GPC) was achieved by transducing 3T3 cells with MSCV vector based on pMIGII (a generous gift of Dr. Jianke Zhang, Thomas Jefferson University) in which the LASV-GPC ORF was amplified by MP3 and MP4 primers (Supplementary Table 1) and added between the EcorI and XhoI restriction digest sites thus generating MSCV-GPC-IRES-GFP. Briefly, MSCV-GPC-IRES-GFP was co-transfected with a pCAGGS-VSV G with Xtreme-Gene 9 in a Gryphon packaging cell line (Allele Biotechnology) and infective retroviral virions were harvested 48 h post transfection. Next, low passage 3T3 murine cell line was transduced with viral supernatant and 8 μg/ml of polybrene and centrifuged at 800xg for 30 min. After 72 h, 3T3 cells were enriched by GFP expression through BD FACSAria II™. Confirmation of LASV GPC expression was done by immunofluorescence by using 50 ul/ml of 9E9 mAb and by FACS using 10 μg/ml of 4C8 mAb. Control target cell line (3T3-MARV) was generated through similar methods but with a Marburg virus GP (Angola strain) expressing MSCV (kind gift from Rohan Keshwara, Thomas Jefferson University).

**Murine NK cell (effector cells) isolation and purification**. Mouse splenocytes obtained from naïve C57BL/6 mouse spleens were made in a single cell suspension through mechanical methods and strained through a 35 μm mesh. Then, the mouse NK Cell Isolation Kit II (MACS-Miltenyi Biotec) was used following the manufacturer's protocol. Purified murine NK cells were collected in RPMI (10% FBS, 50 mM βME, 5 IU/ml of mIL-2 (Biolegend), and 2 ng/ml of mIL-15 (Biolegend) and used immediately for ADCC at either 1:5, 1:10, or 1:20 target to effector cell ratio (see below). Remaining NK cells were stained with 1:200 dilutions of anti-CD3, NK1.1, CD335 (NKp46), CD32/16 markers (BioLegend, catalogue number: 100221; 108709; 137611; 101323), and by 1:1000 dilution of Zombie® UV viability dye (BioLegend) and characterized by flow cytometry (BD LSRFortessa) to confirm NK cell purity and Fcγ-Receptor III expression[29].

**Macrophage effector cells**. IC-21 or J774A.1 macrophages were cultured as per above. At 24 h before an ADCC or ADCP assay, macrophages were scraped in a single cell suspension, centrifuged at 200 g and resuspended in sterile cell culture PBS. For ADCP assays the internal cellular dye CellTrace® Far Red (Invitrogen®) was added following the manufacturer's recommendations. Macrophages were resuspended in serum-free cell culture media containing 5 ng/ml of mGM-CSF (cell signaling technology) and used in the following day for ADCC/ADCP assays and phenotypical analysis. To confirm macrophage phenotype and expression of all Fcγ-receptors[29], macrophages were stained with 1:200 dilution of F4/80, CD64,

CD32/16, and CD16.2 fluorophore-conjugated antibodies (BioLegend, catalogue numbers: 101323; 139303; 123115; 149513) and characterized by flow cytometry, (BD LSRFortessa).

**ADCC/ADCP assays**. Either 1:100 of heat-inactivated sera from immunized mice (see immunizations section), 40 µg/ml of purified IgG from the sera (see immunizations section) or 40 µg/ml of control mAbs (4C8, 9E9 and 5A3) were added to previously seeded $2 \times 10^4$ 3T3-LASV GPC target cells or control target cells and incubated for 30 min at 37 °C in 5% CO2. For Fcγ receptor blockade 100 µg/ml of either anti-CD64, CD32/16, or CD16.2 (BioLegend) was added to effector cells (see above) for 30 min. Next, effector cells were added to target cells at different effector to target cell ratios and incubated for 4 h. Target cells were then dissociated from the plate with Cellstripper® solution (Corning), washed, and resuspended in 200 µl of FACS buffer (5% FBS in PBS) with 30 µg/ml of propidium iodine (PI) viability dye. Cells were then immediately analysed by flow cytometry (BD LSRFortessa).

**ADCP confocal microscopy analysis**. For confocal analysis ADCP assay was conducted in the same conditions as described above but adapted for later microscopy analysis. Briefly, 3T3-LASV GPC target cells were seeded in glass cover-slips and incubated with the respective sera conditions, and then J774A.1 macrophages previously stained with 1:1000 CellTrace® Far Red (see above) were added at a 1:1 Target to effector cell ratio to allow easy visualization. After 4 h, coverslips were washed and mounted in slides with DAPI containing mounting media (VECTASHIELD) and allowed to solidify O/N. Next, day samples were analyzed in a Nikon confocal microscope and further compiled through ImageJ software.

**Gating strategy and ADCC and ADCP analysis**. All flow cytometry data were collected using the FACSDiva (BD) software. Laser voltage settings were adjusted for each analysis by running single color controls. For ADCC analysis, cells were first gated for size using the side scatter (SS) and forward scatter (FS) and selecting the 3T3 population (Supplementary Fig. 3). Next, using the histogram function GFP$^+$ cells were gated and based on this gate a total of 5000 GFP$^+$ events were captured. Due to size variability, ADCP analysis was performed by excluding PI$^+$ events and collecting a total of 10,000 APC$^+$ events (macrophages). For data analysis FlowJo 10 (BD) software was used. The percentage of cytotoxicity (ADCC) was measured by the percentage of PI$^+$ cells of the total GFP$^+$ population after size gating. Since PI is a continuous dye in apoptotic cells[71], PI$^+$ histogram gating was based by defining a 10% PI$^+$ population gate on the control 3T3-LASV GPC cells (no effector cells and sera) as the background. ADCP percentage was measured by measuring the percentage of GFP$^+$/APC$^+$ of the total APC$^+$ population. After defining gating strategy on control cells all gating was applied uniformly to all samples.

**Statistical analysis**. All statistical analysis was performed by using the Graphpad 7 (Prism). To determine the statistical test to be used the population was first analyzed to check whether it followed a normal distribution (Gaussian curve) by applying a D'Agostino-Pearson omnibus normality test. If so a parametric two-tailed T-test was used for comparison within two groups. For grouped analysis, a one-way ANOVA or two-way ANOVA test was used and a post-Hoc analysis using either Sidak or Tukey Honest significant Difference Test with a 95% confidence interval to test significance within groups. Non-parametric tests were used if the population did not follow a normal distribution (indicated in the figure legends).

## Data availability

All relevant data are available from the corresponding author upon request. Sequences of LASV GPC are available at Genbank under accession number MH778559.

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

## Acknowledgements

This work was supported in part by NIH grants R01 AI105204 to M.J.S., by the Jefferson Vaccine Center, and by the Fundação para a Ciência e Tecnologia (FCT) scholarship PD/BD/105847/2014 (to T.A.-M.). This work was also funded in part through the NIAID Division of Intramural Research and the NIAID Division of Clinical Research, Battelle Memorial Institute's prime contract with the U.S. National Institute of Allergy and Infectious Diseases (NIAID) under Contract No. HHSN272200700016I. K.R.H. performed this work as an employee of Battelle Memorial Institute. K.C., an employee of Charles River Laboratories performed this work as a subcontractor to Battelle Memorial Institute. We thank Jennifer Wilson (Thomas Jefferson University, Philadelphia, PA) for critical reading and editing of the manuscript and Jiro Wada for help with the preparations of the illustrations.

## Author contributions

T.A.-M. designed and performed experiments, analyzed data, and wrote the paper; K.R.H, K.C., G.T., and C.W. designed and performed experiments; R.F.J. and M.J.S. designed experiments, analyzed data, and co-wrote the paper. P.B.J. analyzed data and the edited paper.

## Additional information

**Competing interests:** T.A.-M, P.B.J., and M.J.S. are inventors on the U.S. Provisional Patent Application No. 62/691,413 (Title: Non-neutralizing antibodies elicited by recombinant Lassa–Rabies vaccine are critical for protection against Lassa fever). All remaining authors declare no competing interests.

