## [Peer Review File · Nature Communications]

Reviewers' comments:

Reviewer #1 (Remarks to the Author):

The authors present the development and characterization of an inactivated recombinant LASV and rabies vaccine candidate (LASSARAB) expressing a codon-optimized LASV glycoprotein (coGPC). When adjuvanted with a TLR-4 agonist LASSARAB elicited a humoral response against LASV and RABV in both mice and guinea pigs, and protected against LASV challenge. Important evidence for non-neutralizing LASV GPC specific antibodies as a mechanism of protection against Lassa fever is presented for the first time. This is an extremely well written presentation of a complex series of studies. The recent increase in Lassa fever vaccine initiatives by CEPI and others make these results extremely timely and important.

The fact that the immunization induced high titers against Rabies virus glycoprotein, but not LASV GPC confirms that native GPC is not a good inducer of neutralizing antibodies. Hastie et al. Science 02 Jun 2017: Vol. 356, Issue 6341, pp. 923-928 suggest that this is because the most potent neutralizing antibodies recognize quarternary epitopes. This paper should be referenced and discussed in the context of the authors' current results.

Minor comments:

1. ABSTRACT

"Overall, these findings are the first to demonstrate an effective inactivated LF vaccine and elucidate novel humoral correlate of protection for LF." Should be: "Overall, these findings demonstrate an effective inactivated LF vaccine and elucidate novel humoral correlate of protection for LF."

While one dose of inactivated LASV did not protect NHPs [McCormick1993 Journal of Medical Virology 37(1):1-7, in another study 3 doses did protect [Krasnianskii 1993 Vopr. Virusol. 38(6), 276-279].

2. "Indeed, proving the urgency and importance of preventive measures, an unprecedented major LF outbreak, with 25.4% high case fatality rate, is currently unfolding in the major city of Lagos and other parts of Nigeria"

Revise: this was a surge not an outbreak as cases occur in Nigeria yearly. While cases were present in Lagos state, the vast majority were not.

3. P3 The genus *Mastomys* should be capitalized *Mastomys*

4. P8 referred to referred.

5. P17 non-humane to non-human

6. P17 "As a major LF outbreak unfolds in Nigeria at the time of manuscript preparation..." Again, a surge not an outbreak.

7. Good job overall with color selection, but may want to revise Figures 4- 6 to be more color-blind friendly.

Reviewer #2 (Remarks to the Author):

General remarks

This is an interesting study investigating novel vaccine candidates against the highly pathogenic Lassa virus (LASV). The Old World arenavirus LASV is the causative agent of a severe hemorrhagic fever with high mortality in humans and represents one of the most important emerging pathogens according to the World Health Organization. Despite significant efforts over the past decades, the development of a safe and efficacious vaccine against LASV remains an unmet challenge. A particular problem in LASV vaccine design is the notoriously low immunogenicity of LASV envelope glycoprotein (GP)-1 implicated in receptor binding and target for neutralizing antibodies. During natural infection, anti-viral CD8T cells represent a major correlate of protection, whereas neutralizing antibodies (nAb) appear late in convalescence and tend to be of low titer. An elegant recent study by Pinschewer and colleagues (reference 45) provided evidence for N-glycan shielding as a likely mechanism behind low immunogenicity of LASV GP1, similar to envelope GPs of other viruses, such as HIV-1. Recent proof-of-concept studies using recombinant nAb against LASV provided evidence for protection, but the extent of protection did not always correlate with nAb titers.

In the present study, Abreu-Mota and colleagues designed bivalent LASV rabies virus (RABV) vaccines based on expressing LASV GPC within a well-characterized RABV vaccine backbones with favorable safety profile (LAASARAB). Using an existing recombinant LASV candidate vaccine based on recombinant vesicular stomatitis virus (rVSV-LASVGP) as benchmark, the authors first studied the humoral immune response to LASSARAB in a murine model. Interestingly, immunization with live LASSARAB and a variant lacking the RABV G protein failed to induce potent antibody responses. However, inactivated vaccine formulations of LASSARAB, in particular when combined with a TLR4 agonist adjuvant and applied in a prime-boost regimen induced robust titers anti-LASVGP IgG. The newly developed ELISA involved purified rVSV-LASVGP as antigen, allowing the detection of antibodies capable of binding the LASV GP in its native pre-fusion conformation, as displayed on virions and at the surface of infected cells, which is a decisive advantage. Using a state-of-the-art neutralization assay based on VSV pseudotypes, only negligible nAb titers were detected, suggesting that virus-binding, non-nAb may contribute to protection, which is a reasonable assumption. As a LASV challenge paradigm the guinea pig model was used, which is appropriate and widely applied for evaluation of drugs and vaccines against LASV prior to NHP studies. Inactivated LASSARAB combined with adjuvant in a prime-boost application conferred a level of protection comparable to rVSV-LASVGP, which is one of the most promising live vaccine candidates. Consistent with the evaluation of the vaccine in mice, survival did not correlate with nAb titers, which were negligible. Using a combination

of elegant assays, the authors provide evidence for enhanced effector cell activation and clearance mediated by the non-nAb in survivors implicating antibody-dependent cellular cytotoxicity (ADCC) and cellular phagocytosis (ADCP).

The study addresses an important and timely issue in vaccine design against a major emerging pathogen. Although not conceptually novel, the design of a bivalent RABV/LASV vaccine for application in Western Africa is of interest. The study is well-conceived, developed in a logical manner, and is overall well performed. A particular strength is the application of a novel ELISA allowing the detection and quantification of virus-binding non-nAb. A decisive advantage of the inactivated LASSARAB vaccine formulation developed here is its capacity to elicit protective immunity without biosafety concerns in immunocompromised individuals. This is of utmost importance given the population composition in affected regions. The results will contribute to improve rationale LASV vaccine designs. However, some points should be addressed prior to publication.

Specific comments:

Major points:

1. In Fig. 2a, the authors employ immunofluorescence to assess the expression of LASVGP by the different vaccine platforms. It would be better to use flow cytometry to quantify expression of the GPs and to show co-expression of LASV GP and RABV G.
2. A hallmark of LASV GP1 is its dense glycan shield that correlates with its poor immunogenicity (reference 45) and fully glycosylated LASV GP shows an apparent molecular mass of 40-45 kDa in SDS-PAGE. In Fig. 2c the apparent mass of LASV GP1/GP2 seems < 40kDa. Please comment. Also, previous studies revealed the presence of mainly high-mannose sugars on LASV GP1 (Illick, M.M., Branco, L.M., Fair, J.N., Illick, K.A., Matschiner, A., Schoepp, R., Garry, R.F., Guttieri, M.C., 2008. Uncoupling GP1 and GP2 expression in the Lassa virus glycoprotein complex: implications for GP1 ectodomain shedding. *Virology* 5, 161). Have the authors looked at the type of N-glycans present on LASV GP expressed by their LASSARAB vaccine?
3. While strong evidence is provided for a role of non-nAb, and likely ADCC and ADCP in protection, it would be important to better define the actual immunological correlate of protection. It would be very interesting to perform serum transfer and ideally transfer of purified IgG, in the guinea pig challenge model, allowing an assessment of protective non-nAb titers.
4. Has complement-dependence of antibody neutralization been checked?
5. The potent anti-LASV GP antibody response upon vaccination with inactivated LAASARAB (Fig. 4 and S2) is in stark contrast to the low antibody titers in mice immunized with live, replicating virus. It would be interesting to look at the frequency of antigen-specific CD4T cells and B cells in the two situations.

6. In the LASV challenge model, inactivated LASSARAB is compared to live rVSV-LASVGP used as benchmark. Have the authors compared inactivated LASSARAB and inactivated rVSV-LASVGP in combination with GLA-SE?

7. The detection of viral RNA 50 days post challenge (Fig. 6d) is intriguing. Have the authors tried to recover infectious virus from serum or organs of these animals? Do immune-privileged sites (testis, kidney etc.) harbor infectious virus and serve as reservoirs?

Minor points:

1. Introduction, line 19: in the clinic, the SOC for LASV is the off-label use of ribavirin, which shows some efficacy when used early in disease, but can be associated with unwanted side effects, e.g. hemolytic anemia.

2. Line 122: one reason for the more robust growth of LAASARAB lacking the RABV G may be that the GPs interfere during entry due to different fusion pHs and the dependence of LASV GP on LAMP1 for fusion.

3. Please complete the legend for Fig. S2 labelling the curves and be consistent in the nomenclature of live vs. inactivated virus between main text and figures.

4. The robust levels of virus-binding non-nAb elicited by the LASSARAB + GLA-SE formulation reported here are in line with a recent study that reported a similar increase in virus-binding non-nAb to LASV GP1 using a polymersome nanocarrier in combination with the TLR4 agonist MPLA as adjuvant (Galan-Navarro C, Rincon-Restrepo M, Zimmer G, Ollmann Saphire E, Hubbell JA, Hirosue S, Swartz MA, Kunz S. 2017. Oxidation-sensitive polymersomes as vaccine nanocarriers enhance humoral responses against Lassa virus envelope glycoprotein. *Virology* 512:161-171). Please discuss.

Reviewer #3 (Remarks to the Author):

Abreu-Mota present a nice study of a novel Lassa fever vaccine and provide interesting mechanistic studies. The studies are generally well done and should be of interest to Nature Comms readers.

Comments

Fig 1 - rVSV-GPC is a little confusing – the N, P, M, L genes are portrayed as being identical to the Rabies vector but I gather they are VSV genes? - they could be portrayed as a different color (or size?). Or is the whole rabies gene block inserted? – in which case flanking VSV or rabies genes could be illustrated to make this clear.

I gather the BNSP333 vector is similar to a vector used for animal vaccination against rabies

(line 93) – has BNSP333 been used either as a widespread animal vaccine or used in human trials? – if not are there issues of concern that would make this pathway difficult? – this would lessen the enthusiasm for the work as it could make it another interesting but non-translatable vector. I note the related Ebola studies were published many years ago.

Fig 3a - Was the rVSV-GPC group worse than the control rVSV-EGFP group? - why would this be so?

Fig 3c - The high pathogenicity of BNSP333 in suckling mice – is this a problem?

Fig 4 and text – the difference between the replication competent vectors and “inactivated” vectors is not clear. The terminology “referred from now on ” in the text is confusingly different to the Figures.

It is not clear why the unadjuvanted inactivated vector would be better than the live vector, assuming the “dose” is equivalent – was 10ug of the live vector given? Why was only the un-activated vector given 3 times? (this is also a concern for the later protection experiments)

Line 277 the statement “Furthermore, we hypothesize that ADCC might be epitope-dependent given that 3 different mouse LASV GPC-specific mAbs did not induce ADCC killing above background in contrast with the sera or purified IgG (not shown).” – some more context for this statement is needed – to which regions did they bind and did the Mabs have the same Fc isotype? Have Fc-defective (GLGR) mutations been studied for these mabs – that would provide more definitive evidence in my view.

The numbers of animals in Fig 8b are small for unclear reasons. The vaccinated Fcg knock out animals look almost worse than controls – if confirmed this might suggest a role for Fc-mediated function in partial post infection control in this admittedly contrived model, as noted in the discussion.

Mouse and human FcγR are different and there is some scepticism about the Fcg knock out model used – alternate models, including knock in models, have been studied. This could be noted in the discussion.

Minor comment

Line 37. You could note this is also known as the common African rat.

We want to thank the reviewer for their thoughtful evaluation of our manuscript and address their comments below highlighted in blue.

Reviewer #1 (Remarks to the Author):

The authors present the development and characterization of an inactivated recombinant LASV and rabies vaccine candidate (LASSARAB) expressing a codon-optimized LASV glycoprotein (coGPC). When adjuvanted with a TLR-4 agonist LASSARAB elicited a humoral response against LASV and RABV in both mice and guinea pigs, and protected against LASV challenge. Important evidence for non-neutralizing LASV GPC specific antibodies as a mechanism of protection against Lassa fever is presented for the first time. This is an extremely well written presentation of a complex series of studies. The recent increase in Lassa fever vaccine initiatives by CEPI and others make these results extremely timely and important.

The fact that the immunization induced high titers against Rabies virus glycoprotein, but not LASV GPC confirms that native GPC is not a good inducer of neutralizing antibodies. Hastie et al. Science 02 Jun 2017: Vol. 356, Issue 6341, pp. 923-928 suggest that this is because the most potent neutralizing antibodies recognize quarternary epitopes. This paper should be referenced and discussed in the context of the authors' current results.

The paper has been added and we discuss the related findings in the discussion section.

Minor comments:

1. ABSTRACT

“Overall, these findings are the first to demonstrate an effective inactivated LF vaccine and elucidate novel humoral correlate of protection for LF.” Should be: “Overall, these findings demonstrate an effective inactivated LF vaccine and elucidate novel humoral correlate of protection for LF.”

While one dose of inactivated LASV did not protect NHPs [McCormick 1993 Journal of Medical Virology 37(1):1-7, in another study 3 doses did protect [Krasnianskii 1993 Vopr. Virusol. 38(6), 276–279].

We agree that with this suggestion and changed the abstract accordingly.

2. “Indeed, proving the urgency and importance of preventive measures, an unprecedented major LF outbreak, with 25.4% high case fatality rate, is currently unfolding in the major city of Lagos and other parts of Nigeria”

Revise: this was a surge not an outbreak as cases occur in Nigeria yearly. While cases were present in Lagos state, the vast majority were not.

We agree with this assessment and corrected this within the text

3. P3 The genus mastomys should be capitalized Mastomys

We did correct this as suggested.

4. P8 referred to referred.

Has been corrected accordingly.

5. P17 non-humane to non-human

Has been corrected accordingly.

6. P17 “As a major LF outbreak unfolds in Nigeria at the time of manuscript preparation...” Again, a surge not an outbreak.

Has been corrected accordingly.

7. Good job overall with color selection, but may want to revise Figures 4- 6 to be more color-blind friendly.

We adjusted all figures so they are color-blind friendly.

Reviewer #2 (Remarks to the Author):

General remarks

This is an interesting study investigating novel vaccine candidates against the highly pathogenic Lassa virus (LASV). The Old-World arenavirus LASV is the causative agent of a severe hemorrhagic fever with high mortality in humans and represents one of the most important emerging pathogens according to the World Health Organization. Despite significant efforts over the past decades, the development of a safe and efficacious vaccine against LASV remains an unmet challenge. A particular problem in LASV vaccine design is the notoriously low immunogenicity of LASV envelope glycoprotein (GP)-1 implicated in receptor binding and target for neutralizing antibodies. During natural infection, anti-viral CD8T cells represent a major correlate of protection, whereas neutralizing antibodies (nAb) appear late in convalescence and tend to be of low titer. An elegant recent study by Pinschewer and colleagues (reference 45) provided evidence for N-glycan shielding as a likely mechanism behind low immunogenicity of LASV GP1, similar to envelope GPs of other viruses, such as HIV-1. Recent proof-of-concept studies using recombinant nAb against LASV provided evidence for protection, but the extent of protection did not always correlate with nAb titers.

In the present study, Abreu-Mota and colleagues designed bivalent LASV rabies virus (RABV) vaccines based on expressing LASV GPC within a well-characterized RABV vaccine backbones with favorable safety profile (LAASARAB). Using an existing recombinant LASV candidate vaccine based on recombinant vesicular stomatitis virus (rVSV-LASVGP) as benchmark, the authors first studied the humoral immune response

to LASSARAB in a murine model. Interestingly, immunization with live LASSARAB and a variant lacking the RABV G protein failed to induce potent antibody responses. However, inactivated vaccine formulations of LASSARAB, in particular when combined with a TLR4 agonist adjuvant and applied in a prime-boost regimen induced robust titers anti-LASVGP IgG. The newly developed ELISA involved purified rVSV-LASVGP as antigen, allowing the detection of antibodies capable of binding the LASV GP in its native pre-fusion conformation, as displayed on virions and at the surface of infected cells, which is a decisive advantage. Using a state-of-the-art neutralization assay based on VSV pseudotypes, only negligible nAb titers were detected, suggesting that virus-binding, non-nAb may contribute to protection, which is a reasonable assumption. As a LASV challenge paradigm the guinea pig model was used, which is appropriate and widely applied for evaluation of drugs and vaccines against LASV prior to NHP studies. Inactivated LASSARAB combined with adjuvant in a prime-boost application conferred a level of protection comparable to rVSV-LASVGP, which is one of the most promising live vaccine candidates. Consistent with the evaluation of the vaccine in mice, survival did not correlate with nAb titers, which were negligible. Using a combination of elegant assays, the authors provide evidence for enhanced effector cell activation and clearance mediated by the non-nAb in survivors implicating antibody-dependent cellular cytotoxicity (ADCC) and cellular phagocytosis (ADCP).

The study addresses an important and timely issue in vaccine design against a major emerging pathogen. Although not conceptually novel, the design of a bivalent RABV/LASV vaccine for application in Western Africa is of interest. The study is well-conceived, developed in a logical manner, and is overall well performed. A particular strength is the application of a novel ELISA allowing the detection and quantification of virus-binding non-nAb. A decisive advantage of the inactivated LASSARAB vaccine formulation developed here is its capacity to elicit protective immunity without biosafety concerns in immunocompromised individuals. This is of utmost importance given the population composition in affected regions. The results will contribute to improve rationale LASV vaccine designs. However, some points should be addressed prior to publication.

We want to thank reviewer two for her/his vigorous evaluation and supporting the importance of the study. We agree with several of the concerns raised and addressed them below.

Specific comments:

Major points:

1. In Fig. 2a, the authors employ immunofluorescence to assess the expression of LASVGP by the different vaccine platforms. It would be better to use flow cytometry to quantify expression of the GPs and to show co-expression of LASV GP and RABV G.

That is a reasonable request and the flow cytometry data has been added to Figure 2.

2. A hallmark of LASV GP1 is its dense glycan shield that correlates with its poor immunogenicity (reference 45) and fully glycosylated LASV GP shows an apparent molecular mass of 40-45 kDa in SDS-PAGE. In Fig. 2c the apparent mass of LASV GP1/GP2 seems < 40kDa. Please comment. Also, previous studies revealed the presence of mainly high-mannose sugars on LASV GP1 (Illick, M.M., Branco, L.M., Fair, J.N., Illick, K.A., Matschiner, A., Schoepp, R., Garry, R.F., Guttieri, M.C., 2008. Uncoupling GP1 and GP2 expression in the Lassa virus glycoprotein complex: implications for GP1 ectodomain shedding. *Virology* 5, 161). Have the authors looked at the type of N-glycans present on LASV GP expressed by their LASSARAB vaccine?

Regarding the first question in point 2, indeed we misidentified LASV GP1 as GP2 in our Western Blots while only LASV GP2 was being detected (as later confirmed by GP2 specific human 22.5D mAb). Unfortunately, as we did not possess a GP1 mAb that detects monomeric GP1 on western blot at the time of writing, we focused on proving the presence of both GP1 and GP2 by ELISA along with the presence of a fully conformational LASV GPC on the virion surface 37.7H mAb (confirmed in ELISA – figure S1g) with previously described mAbs in reference 43.

As such, to provide the information requested we now include a Western blot probed with guinea pig survivor sera that detects monomeric LASV GP1 alongside GP2 and has been included in both figure 2 and figure S1. Indeed, as stated, we can confirm that LASV GP1 is running with molecular size, as previously described (from 48 kDa to 42 kDa), thus suggesting that correct GP1 glycosylation is occurring. To confirm such findings and to corroborate with the results of Illick and Branco&Garry (Characterization of the Lassa virus GP1 ectodomain shedding: implications for improved diagnostic platforms, 2009 *Virology* journal) and more recent works, we treated LASSARAB-inactivated particles with the endoglycosidases Endo H and PNGase F and have now included the results as part of Supplemental figure 1. (S1f) Our results showed a mobility shift for GP1 when treated with Endo H (from 45 kDa to a gradient between 45 kDa to 35 kDa) and when treated with PNGase (further shift to 20-23 kDa), thus indicating the presence of N-Glycans on LASV GPC. (S1e) Similar results are observed with GP2, which results in a mobility shift from 42-38 kDa to 34-30 kDa when treated with Endo H and in a further reduction to 20-23 kDa when treated with PNGase F.

Since RABV P protein runs in a similar size and fuzzy pattern as LASV GP1 it is hard to discriminate GP1 in LASSARAB particles resolved in a SDS-PAGE page although a faint enhancement of signal can be observed (now indicated with an arrow on figure 2d). GP1 becomes notoriously apparent in SDS-PAGE when purified LASV GPC antigen is resolved in SDS-PAGE (now included as Supplemental figure 1b).

The presence of GP1 monomer with a molecular size of 42-48 kDa is also observed in the ELISA antigen together with GP2 (S1 a, b and c) in both western blot and SDS-PAGE, thus indicating that a similar glycosylation pattern is present in LASV GPC antigen used for the detection of GPC binding IgGs.

It should be noted that depending on the cell line used to grow the virus we did observe some slight variation in size and pattern on GPC/GP2 in western blot (data not included but can be provided if requested).

3. While strong evidence is provided for a role of non-nAb, and likely ADCC and ADCP in protection, it would be important to better define the actual immunological correlate of protection. It would be very interesting to perform serum transfer and ideally transfer of purified IgG, in the guinea pig challenge model, allowing an assessment of protective non-nAb titers.

We agree this would be an interesting addition to the paper, it is nevertheless, by itself, a time consuming and expensive experiment (due to the requirement of both a BSL-4 facility and guinea pig/NHP model) that would only complement the results we obtained *in vivo*, in mice, with our surrogate LASV challenge mouse model. Applying our ADCC/ADCP assays for guinea pig sera would also not be possible to be conducted with accuracy since Guinea pig specific reagents (NK/macrophage cell isolation kits) are not available as well as guinea pig IgG sub isotype profiling. Moreover, while this experiment could confirm that passive sera transfer of antibody from immunized guinea pigs is sufficient for protection in the guinea pig model, it can also introduce a confounding factor. Since LASSARAB immunization also induces CD8+ and CD4+ T cellular responses those controls would also be required if a definitive protective non-nAb titer induced by LASSARAB is to be established.

These concerns formed the rationale behind the experiment with Fcγ-R KO mice (Figure 8). With this experiment, we sought to emulate more closely what would happen if, after a LASSARAB immunization, mice were exposed to a virus with a similar tropism as LASV (since it expresses the same glycoprotein) but in the absence of Fc receptor functions.

4. Has complement-dependence of antibody neutralization been checked?

That is a very good suggestion. Indeed, we had checked the complement-dependence of antibody neutralization while developing the VNA assay and found that there seemed to be no function for complement-mediated neutralization of LASV pseudotypes by antibodies elicited by our vaccine.

We had not included these results since they were negative and served as a basis for developing the VNA assay, but we have now included them as part of supplemental figure S3 since other researchers might have the same question.

5. The potent anti-LASV GP antibody response upon vaccination with inactivated LAASARAB (Fig. 4 and S2) is in stark contrast to the low antibody titers in mice immunized with live, replicating virus. It would be interesting to look at the frequency of antigen-specific CD4T cells and B cells in the two situations.

The potent response is indeed interesting, and we see have seen a similar response for some other viral antigens (e.g., MERS-CoV) but not for all (e.g., Nipah). To a certain extent, the response seems to go against the dogma that live vaccines are more potent than killed. The overall hypothesis is that the immune response against RABV G might quickly block the spread of the vector (at least in intramuscular immunization) and therefore prevent a potent IgG immune response against LASV GPC. However, RABV G immune response was not compromised, as shown in Figure S2b, to a certain extent corroborating previous findings that LASV GPC is a very poor immunogen by itself (Ref: 43, 81 and 82).

In the case of the RABV G deleted vector, we saw an increase in GPC response compared to the RABV G-containing vector, but adjuvanted killed vaccine was still more potent thereby indicating that RABV vector is being cleared before inducing a significant immunological response to LASV GPC.

Because of the poor response of the replication competent rabies vectors and higher advantages of inactivated vaccine, this study was then largely directed toward the development of a killed vaccine and no further experiments were conducted with live (replication competent) rabies vectors.

6. In the LASV challenge model, inactivated LASSARAB is compared to live rVSV-LASVGP used as benchmark. Have the authors compared inactivated LASSARAB and inactivated rVSV-LASVGP in combination with GLA-SE?

We included the VSV vaccine in the form it is currently utilized because it is a leading vaccine candidate. To study VSV in its inactivated form would certainly be very interesting, but nevertheless it was outside the scope of this work. Of note, a VSV-based inactivated vaccine would not confer protection to RABV which would be surely a major disadvantage for the intended region.

7. The detection of viral RNA 50 days post challenge (Fig. 6d) is intriguing. Have the authors tried to recover infectious virus from serum or organs of these animals? Do immune-privileged sites (testis, kidney etc.) harbor infectious virus and serve as reservoirs?

We were equally surprised by the detection of viral RNA 50 days post challenge. A decisive factor for doing such a prolonged monitoring post challenge was that we observed late clinical symptoms (day 20 post challenge), and there is very little information in the literature regarding persistent infection of LASV. As such, the persistence of the LASV infection was an unforeseen result that will be addressed in a following work since recently both (40) and (42) had similar findings albeit in an NHP model and in a shorter timeframe. Our collaborators at the IRF are currently studying the persistence of the virus at different sites.

Minor points:

All of reviewer two's minor points listed below have merit, and therefore we did change the text of the manuscript accordingly.

1. Introduction, like 19: in the clinic, the SOC for LASV is the off-label use of ribavirin, which shows some efficacy when used early in disease, but can be associated with unwanted side effects, e.g. hemolytic anemia.
2. Line 122: one reason for the more robust growth of LAASARAB lacking the RABV G may be that the GPs interfere during entry due to different fusion pHs and the dependence of LASV GP on LAMP1 for fusion.
3. Please complete the legend for Fig. S2 labelling the curves and be consistent in the nomenclature of live vs. inactivated virus between main text and figures.
4. The robust levels of virus-binding non-nAb elicited by the LASSARAB + GLA-SE formulation reported here are in line with a recent study that reported a similar increase in virus-binding non-nAb to LASV GP1 using a polymersome nanocarrier in combination with the TLR4 agonist MPLA as adjuvant (Galan-Navarro C, Rincon-Restrepo M, Zimmer G, Ollmann Saphire E, Hubbell JA, Hirosue S, Swartz MA, Kunz S. 2017. Oxidation-sensitive polymersomes as vaccine nanocarriers enhance humoral responses against Lassa virus envelope glycoprotein. *Virology* 512:161-171). Please discuss.

Reviewer #3 (Remarks to the Author):

Abreu-Mota present a nice study of a novel Lassa fever vaccine and provide interesting mechanistic studies. The studies are generally well done and should be of interest to Nature Comms readers.

Comments

Fig 1 - rVSV-GPC is a little confusing – the N, P, M, L genes are portrayed as being identical to the Rabies vector but I gather they are VSV genes? - they could be portrayed as a different color (or size?). Or is the whole rabies gene block inserted? – in which case flanking VSV or rabies genes could be illustrated to make this clear.

Yes, they are VSV genes. We will change the figure accordingly to avoid such confusion

I gather the BNSP333 vector is similar to a vector used for animal vaccination against rabies (line 93) – has BNSP333 been used either as a widespread animal vaccine or used in human trials? – if not are there issues of concern that would make this pathway difficult? – this would lessen the enthusiasm for the work as it could make it another

interesting but non-translatable vector. I note the related Ebola studies were published many years ago.

We give some background regarding the vector below but the most important fact here is that we do develop a deactivated (killed) not a live rabies virus-based vaccine against LASV. This is very important because pregnant women and children are a major target of the vaccine. For certain other approaches we consider immunization of animal hosts with the live virus, but not humans. This is now clearly stated in the publication.

The vector:

The BNSP333 vector is based on the SAD-B19 vaccine strain of rabies virus. This vaccine strain has been widely used for live oral immunization since the 70s, as can be seen in here: http://www.who.int/rabies/vaccines/oral_immunization/en/.

BNSP333 was further attenuated by a 333 mutation in its native glycoprotein that completely abrogates neurovirulence even in SCID (severely immunocompromised mice), as can be observed in our works as well. Furthermore, this vector is permissible to recovery in GLP/GMP conditions.

Currently, 3 RABV-based vaccines based on BNSP333 are being manufactured and formulated using good manufacturing practices (GMP). There are all utilized as deactivated vaccines, so the attenuation of the vectors is mostly an advantage for production. Together with LASSARAB (also based on BNSP333 platform), they are the basis for the tetravalent vaccine development NIH contract (HHSN272201700082C) against EBOV, SUDAV, MARV and LASV and are on target to reach human clinical trials.

Several studies using BNSP333 for several different diseases (Hendra, Ebola and MERS) have been published within the last few years (as recent as 2017) and can be verified here in the paper's literature: (54-61)

Fig 3a - Was the rVSV-GPC group worse than the control rVSV-EGFP group? - why would this be so?

Yes this is true - rVSV-EGFP has an extra gene (GFP) and contains the original VSV glycoprotein, hence has likely a different tropism. Therefore VSV-GPC seems to be neurotropic. It is however safe if used in peripheral inoculations for the purpose of immunization in immunocompetent mice/guinea pigs.

Fig 3c - The high pathogenicity of BNSP333 in suckling mice – is this a problem?

Our work presents an intracranial exposure of virus in highly susceptible mice that do not possess a fully developed immune system. The high pathogenicity was expected, and the main objective was to confirm that the addition of LASV GPC to BNSP333 did not augment killing.

We also conducted the same experiment in the highly susceptible adult SCID mice, and as it can be confirmed, a mature innate immune response in the absence of adaptive immune response is sufficient to clear the virus even after an intracranial exposure. As such, the virus should not be of concern to even severely immunocompromised people (if used as replication competent).

Furthermore, and as described above, the objective is to use the vaccine as an inactivated vector and, as such, no live virus will actually be exposed to humans.

Fig 4 and text – the difference between the replication competent vectors and “inactivated” vectors is not clear. The terminology “referred from now on” in the text is confusingly different to the Figures.

This has been corrected.

It is not clear why the unadjuvanted inactivated vector would be better than the live vector, assuming the “dose” is equivalent – was 10 µg of the live vector given? Why was only the un-activated vector given 3 times? (this is also a concern for the later protection experiments)

The dose equivalent of live/dead vaccine is not a straightforward concept since live vaccines replicate in the tissues and killed do not, and as such antigenic exposure is different depending on tissue/virus/immunogen. As such, it is more appropriate to measure a live vaccine dosage as a measure of live infectious units. Of note, the dosage equivalent of live infectious units (based on total particles) used as a deactivated vaccine, would likely be far too low to be effective (since it won't replicate). Thus inactivated vaccines to be effective need be used at higher particle concentrations than replication competent equivalent.

The inactivated vaccine was given 3 times (and, later on, switched to 2 times [on Day 0 and Day 28] since the Day 7 boost does not seem to make a difference) to account for the lack of replication of an inactivated vector, and also to follow the typical rabies virus immunization schedule. The live vectors were not boosted because we previously showed that this approach does not work for replication competent RABV. Also, by using inactivated vaccine, safety concerns are greatly reduced and boosts can be employed more easily.

Line 277 the statement “Furthermore, we hypothesize that ADCC might be epitope-dependent given that 3 different mouse LASV GPC-specific mAbs did not induce ADCC killing above background in contrast with the sera or purified IgG (not shown).” – some more context for this statement is needed – to which regions did they bind and did the mAbs have the same Fc isotype? Have Fc-defective (GLGR) mutations been studied for

these mabs – that would provide more definitive evidence in my view.

Our main objective with this experiment was to prove that ADCC was dependent on IgG, and, to a further extent, verify if any GPC specific mAbs (that are IgG2a/b subclass) can induce ADCC, which was not the case. We will retract that statement, since the mAbs used in that experiment are currently being characterized, the epitope dependence of ADCC in LASV will be published as a separate work.

The numbers of animals in Fig 8b are small for unclear reasons. The vaccinated Fcγ knock out animals look almost worse than controls – if confirmed this might suggest a role for Fc-mediated function in partial post infection control in this admittedly contrived model, as noted in the discussion.

We have now edited the text and the figure to better reflect the numbers of animals per group since the animal numbers are not small (10 per group in the LASSARAB vaccinated animals so 20 in total which was the main experiment). For controls we had 5 per group (so 10 in total). Moreover, these numbers were large enough to be statistically significant

Mouse and human FcγR are different and there is some scepticism about the Fcγ knock out model used – alternate models, including knock in models, have been studied. This could be noted in the discussion.

That is a very good point and shall be fully noted in the discussion.

Minor comment

Line 37. You could note this is also known as the common African rat.

True and addressed.

REVIEWERS' COMMENTS:

Reviewer #2 (Remarks to the Author):

The authors have in my opinion addressed the major points of criticism in a satisfactory manner and I have no further comments.

Reviewer #3 (Remarks to the Author):

I am satisfied with the response.